# How to Build a Pre-trained Multimodal model for Simultaneously Chatting and Decision-making?

## Abstract

Existing large pre-trained models typically map text input to text output in an end-to-end manner, such as ChatGPT, or map a segment of text input to a hierarchy of action decisions, such as OpenVLA. However, humans can simultaneously generate text and actions when receiving specific input signals. For example, a driver can make precise driving decisions while conversing with a friend in the passenger seat. Motivated by this observation, we consider the following question in this work: is it possible to construct a pre-trained model that can provide both language interaction and precise decision-making capabilities in dynamic open scenarios. We provide a definitive answer to this question by developing a new model architecture termed Visual Language Action model for Chatting and Decision Making (VLA4CD), and further demonstrating its performance in challenging automonous driving tasks. We build VLA4CD on the basis of transformer-based LLM architecture. Specifically, we leverage LoRA to fine-tune a pre-trained LLM with data of multiple modalities covering language, visual, and action. Unlike the existing LoRA operations used for LLM fine-tuning, we have designed new computational modules and training cost functions for VLA4CD. These designs enable VLA4CD to provide continuous-valued action decisions while outputting text responses. In contrast, existing LLMs can only output text responses, and current VLA models can only output action decisions. Moreover, these VLA models handle action data by discretizing and then tokenizing the discretized actions, a method unsuitable for complex decision-making tasks involving high-dimensional continuous-valued action vectors, such as autonomous driving. The extensive experimental results on the closed-loop autonomous driving platform CARLA validate that: (1) the model construction method we proposed is effective; (2) compared to the state-of-the-art VLA model, VLA4CD can provide more accurate real-time decision-making while retaining the text interaction capability inherent to LLMs.

## 1 Introduction

Since the emergence of ChatGPT, large-scale pre-trained models, represented by large language models (LLMs), have garnered increasing attention. LLMs are trained on vast amounts of text and code data on the internet, encoding a significant amount of general knowledge about the real world. This equips them with better generalization capabilities compared to traditional AI models, such as in-context learning abilities and certain reasoning capabilities (through techniques such as chain-of-thought (Wei et al., 2022)). A development trend in the field of large-scale pre-trained models is that their application domains are expanding from tasks like dialogue and text generation to decision-making tasks in the open physical world.

How to build large-scale pre-trained models for decision-making tasks in the open physical world? Currently, there are three major approaches. An approach is to serialize the decision-making process and then train a sequence model, such as the decision transformer (Chen et al., 2021), in the same way as processing text. This method relies on the construction of large-scale high-quality decision-making datasets. The second approach involves adopting a hierarchical modular system design, where the pre-trained LLM provides high-level planning, such as breaking down the target

task into a series of subtasks and then completing each subtask by calling tools or small models aimed at the subtasks (Chen et al., 2024; Carta et al., 2023; Hu et al., 2024; Zhou et al., 2024). This approach requires manual pre-design of system modularization and the establishment of interfaces between modules. Additionally, after the model is deployed, in addition to the latency caused by LLM inference, it also introduces the working latency of other modules, making it unsuitable for decision-making scenarios with high-time requirements. The last approach is to train a multimodal visual language action model (VLA) based on LLM (Padalkar et al., 2023; Kim et al., 2024). Unlike the hierarchical modular method, the VLA model can provide end-to-end decision generation, eliminating the need for manual module design and interface design between modules.

To the best of our knowledge, existing LLM or VLA models, given an input signal (a piece of text prompt, an image, or a video), produce outputs that are single-modal (a piece of text or an action decision). However, we know that for us humans, we can simultaneously generate text and actions when receiving specific input signals. For example, a driver can make precise driving decisions while conversing with a friend in the passenger seat. Inspired by the above observations, we attempt to answer the following question in this paper:

***Is it possible to develop a pre-trained model that can provide both action decision-making and text interaction capabilities in an end-to-end manner?***

We provide a definitive answer to it by developing a new model architecture termed Visual Language Action model for Chatting and Decision Making (VLA4CD), and further demonstrating its performance in challenging automonous driving tasks. Like existing VLA models, VLA4CD is a multimodal pre-trained large model developed based on the transformer architecture. However, it has significant differences from current VLAs (such as RT-X (Brohan et al., 2022; 2023)):

- The operational mechanism of VLA involves executing serialized decisions after receiving text instructions, without generating text data during the decision-making process. In contrast, VLA4CD allows for the synchronous generation of text data during real-time decision-making.
- Current VLA models typically handle action data by discretizing it and then tokenizing the discrete values. This approach is not suitable for complex decision-making scenarios such as automonous driving, where actions are high-dimensional continuous value vectors. Our VLA4CD processes actions directly as continuous values, eliminating the need for discretization and making it more suitable for such scenarios.

In summary, the ***main contributions*** of this work are as follows.

- We propose a new problem setting: how to synthesize the capabilities of LLM and VLA using a single model to achieve end-to-end simultaneous action decision-making and chatting with people.
- We present a solution to the aforementioned problem. Specifically, we propose a method for constructing VLA4CD based on pre-trained LLM and have validated the effectiveness and superiority of this method through extensive closed-loop autonomous driving experiments on CARLA (Dosovitskiy et al., 2017). The experimental results show that the resulting VLA4CD model not only outputs more accurate real-time action decisions compared to the SOTA models but also perfectly retains real-time text-based dialogue functionality. Our method combines several experimentally validated ideas: (1) a computational module and cost function term for generating continuous action values; (2) an image reconstruction loss term added in the training cost function to ensure the exploitation of rich information from the visual modality data during text generation and decision-making processes; (3) a label smoothing strategy to maintain dialogue capabilities and enhance decision-making.
- We will open source our model, code, and dataset after the reviewing process.

## 2 RELATED WORK

### 2.1 LLMS FOR DECISION-MAKING

Since the publication of (Brown et al., 2020), generative Pre-trained Transformer (GPT) has become the most popular training paradigm for building LLMs. LLMs represented by GPT-3.5 and

GPT-4 exhibit significantly enhanced zero-shot generalization and reasoning capabilities compared to previous language models (OpenAI, 2023). The release of the open-source LLaMA series models (Touvron et al., 2023a;b) has accelerated the development of LLMs. In (Wei et al., 2022), a general technique to enhance LLM reasoning capabilities, known as chain-of-thought, was proposed. The work in (Yao et al., 2022) proposed ReAct, which uses LLMs to generate reasoning traces and task-specific actions in an interleaved manner, thereby achieving greater synergy between the two. Additionally, recent works have used LLMs as components in building hierarchical modular decision-making agents, where they are only used to generate high-level plans and do not directly generate decisions (Ahn et al., 2022; Fu et al., 2023; Carta et al., 2023; Chen et al., 2024; Xu et al., 2024; Sha et al., 2023; Liu et al., 2023; Hu et al., 2024; Zhou et al., 2024). The VLA4CD model proposed here can be seen as a multimodal GPT model fine-tuned for a downstream application scenario, featured by its capability to simultaneously output action decisions and textual chatting.

## 2.2 VLA MODEL FOR DECISON-MAKING

The VLA model is a type of model designed to handle multimodal input of vision, language, and action to accomplish embodied decision-making tasks. Unlike traditional LLMs that are mostly used for constructing conversational AI represented by ChatGPT, VLA has the ability to generate a control signal for a physical entity, e.g., a Robot, that interacts with the environment. VLA has been widely used for instruction-following tasks, wherein it endows the agent with an ability to understand language instructions, visually perceive the environment, and generate appropriate actions (Huang et al., 2023; Li et al., 2023b; Zhen et al., 2024; Dorka et al.). Compared to deep reinforcement learning (RL) methods, VLA has shown a remarkable performance gain in versatility, flexibility, and generality in complex environments (Padalkar et al., 2023; Brohan et al., 2023; et al, 2024; Team et al., 2024; Li et al., 2023c; Bai et al., 2023; Li et al., 2022; 2023a; Liu et al., 2024; Tan & Bansal, 2019). However, such VLA models represented by RT-X (Padalkar et al., 2023) and OpenVLA (Kim et al., 2024), typically discretize continuous action spaces into fixed intervals. This action discretization raises significant limitations for them to deal with fine-grained continuous actions that are required for capturing nuanced operations necessary for some complex tasks. This issue is particularly pronounced in scenarios that require high precision and real-time responsiveness, such as autonomous driving.

In our VLA4CD, we propose a technique to avoid action discretization in VLA. In addition, we consider a new problem setting that differs from the instruction-following one. In our setting, the agent can chat with a human and make fine-grained decisions simultaneously. We found that if we directly use current VLA models into this setting, they perform unsatisfactorily as they tend to rely more on the text data to generate decisions while neglecting the critical role of visual information.

## 3 METHODOLOGY

In this section, we present how to build VLA4CD in detail, including the model architecture and the training procedure, with a focus on the loss designs in the last output hiddens layer. An overview of VLA4CD is illustrated in Figure 1. To begin with, we present the problem setting of our concern.

### 3.1 PROBLEM SETTING

We consider a multimodal setting similar as (Xiao et al., 2020), wherein, at each time step $t$, upon the agent performs an action $a_t$, the environment returns an observation consisting of both visual and textual modalities, denoted by $\{o_t, w_t\}$. Our objective is to build a generative model $\pi(a_t, \hat{w}_t | o_{t-H}, w_{t-H}, a_{t-H}, ..., o_t, w_t)$, which can generate both high-quality action decisions and text responses, given a sequence of historic observations and actions. Here, $\hat{w}_t$ denotes a text-formed response to the text-formed input $w_t$. If $w_t$ is a question, then $\hat{w}_t$ can be seen as its answer given by our model. $H$ denotes the length of the context.

### 3.2 MODEL ARCHITECTURE

Our model supports three different input modalities: text, image, and numeric vector. We use Llama-7b (Touvron et al., 2023b) as the backbone model, and encode textual inputs by its pre-trained

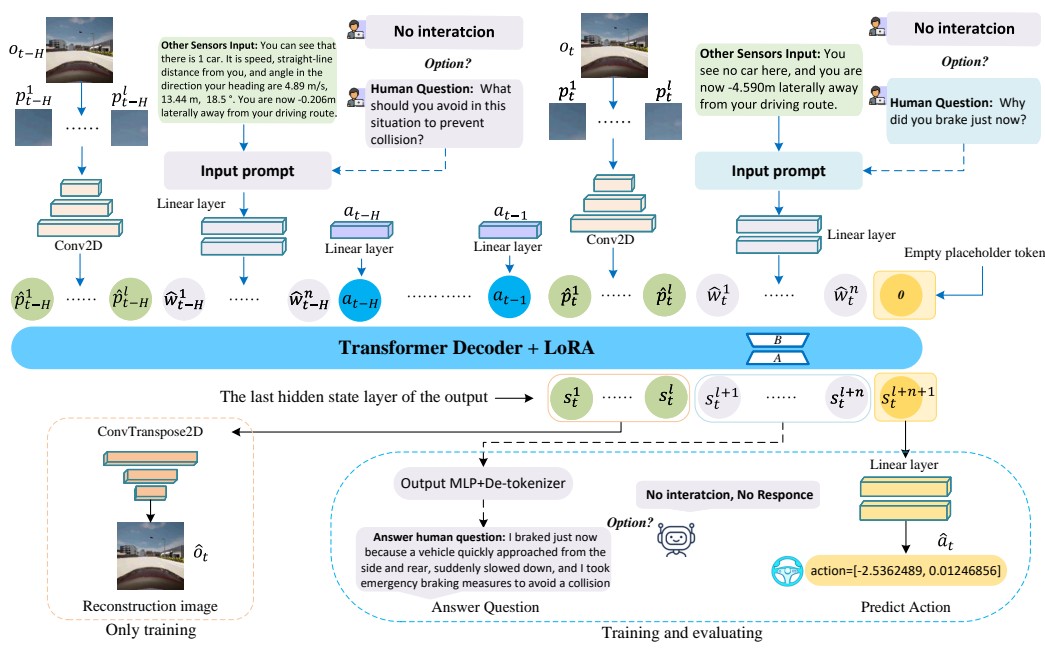

Figure 1: An overview of VLA4CD framework.

embedding layers. To encode the visual inputs, we follow the standard practice used in visual language models (VLMs) (Liu et al., 2024) and VLAs (Kim et al., 2024). Specifically, we first segment each input image $o$ into $L$ patches $p_l, l = 1, \ldots, L,$, then train a 2D convolution network that directly maps the patches to the vector space. In addition, to deal with the input of the action value, we train a multi-layer perceptron (MLP) module that encodes the action values to the vector space. Finally, We concatenate encoded embeddings of all modalities together to form a sequence of embedded trajectory $\tau$ at time $t$ as follows:

$$\tau_t = \{(\hat{p}_{t-H}^1, \ldots, \hat{p}_{t-H}^L), (\hat{w}_{t-H}^1, \ldots, \hat{w}_{t-H}^n), a_{t-H}, \ldots, (\hat{p}_t^1, \ldots, \hat{p}_t^l), (\hat{w}_t^1, \ldots, \hat{w}_t^n)\}, \quad (1)$$

where $\hat{p}_t^i$ and $\hat{w}_t^j$ denote the embeddings of $i$-th patch for visual observation and $j$-th token for textual observation at at time $t$, respectively.

During the inference stage, the transformer backbone in VLA4CD generates the hidden embeddings $s_t^{l+1}, \cdots, s_t^{l+n+1}$ as shown in Figure 1, then these embeddings are decoded into the outputs of different modalities. Specifically, VLA4CD supports two different output modalities: text for chatting and numeric vector for action-level decision making. For the chatting part, we use the pre-trained output MLP layers and tokenizer of the Llama-7b model to generate texts. For action decision-making, our model generates one more embedding vector after the "$< EOS >$", an empty placeholder token. Unlike previous work like OpenVLA (Kim et al., 2024) and RT-X (Brohan et al., 2023), in which action prediction is formalized as a token generation task by splitting the action space into discrete action bins, we train an action head consisting of a two-layer MLP module. This action head directly maps the output embedding to action values. We empirically find that using our approach leads to better performance compared to discretizing action values.

## 3.3 TRAINING PROCEDURE

We fine-tune the transformer backbone with LoRA (Hu et al., 2021) and train the image encoding module, text encoding module, action encoding, and decoding modules with an offline dataset $D_{expert}$, which contains demonstrated trajectories of driving vehicles with question-answer pairs related to this driving scenario. The training objective is to predict accurate actions for vehicle control and answer domain questions such as "*Summarize the current driving scenario at a high level*". Moreover, to encourage the model to abstract key information from the images and prevent over-

fitting, we consider image reconstruction as an auxiliary task, adding a 2D transposed convolution layer to reconstruct input images patches from the output last hidden embeddings $s_t^1, \cdots s_t^l$, as illustrated in Figure 1. As a result, our training loss is composed of three items, corresponding to text generation, action prediction, and image reconstruction, respectively. Next, we describe each loss item in detail. For ease of presentation, we denote the parameters in the auxiliary image decoder as $\phi$ and all other trainable parameters as $\theta$.

**Text Generation**    In our experiment, we found that merely replacing specific numerical values in the translation template (Chen et al., 2024) results in minimal representational differences caused by the sequential nature of the data, making the phenomenon of model overfitting easy to happen if we use the conventional cross-entropy loss for text generation. Refer to Appendix A.7 for details. To mitigate this, we use the label smoothing technique to regularize the training process (Szegedy et al., 2016). Specifically, the hard label for token $w_i$ is smoothed by assigning a small portion of the probability mass to incorrect classes:

$$q_i^k = \begin{cases} 1 - \epsilon & \text{if } k = y_i, \\ \frac{\epsilon}{K-1} & \text{otherwise,} \end{cases} \tag{2}$$

where $\epsilon$ is the smoothing factor and $K$ is the number of total classes, i.e., vocabulary size. That is to say, the loss item for text generation we finally use is:

$$\mathcal{L}_{\text{language}}(\theta) = \frac{1}{N} \sum_i \sum_k q_i^k \log p(k|\tau^{:i-1}, \theta), \tag{3}$$

where $\tau^{:i-1}$ denotes the input token sequence before position $i$, used for predicting token $i$. $N$ denotes the maximum padding length to unify the input text.

**Action Prediction**    To directly predicts continuous action values instead of discrete action bins, we train our model with a mean square error (MSE) loss between the ground-truth action value $a_t$ and the predicted value, as follows:

$$\mathcal{L}_{\text{action}}(\theta) = \frac{1}{T} \sum_t^T \frac{1}{D} \sum_d^D [(a_t^d - \pi(\tau_t, \theta))^2] \tag{4}$$

where $D$ denotes the dimension of the action space. In our experiments, the action dimension is 2, corresponding to the acceleration and steering of the vehicle, respectively.

**Image Reconstruction**    The visual modality data contains rich information about the states of the environment. However, we find that, with a limited dataset, directly training the image encoder from language and action losses is not sufficient, as it leads to information losses. Inspired by Hafner et al. (2019), we consider an auxiliary image reconstruction task to introduce additional supervision in the visual modality. Specifically, we use a 2D transposed convolution layer $f_\phi$ to reconstruct each image patch from its corresponding output embedding and train the model to minimize the pixel-wise Euclidean distance between the original and reconstructed image patches:

$$\mathcal{L}_{\text{image}}(\theta, \phi) = \frac{1}{L} \sum_l^L \text{MSE}(o_t, f_\phi(\pi(g_\theta(\tau_t^{:p_t^l}), \theta))), \tag{5}$$

where $o_t$ is the input image, and $\tau_t^{:p_t^l}$ is the input sequence up to this patch token, and $g_\theta$ represents a trainable 2D convolutional network that directly maps image patches $p_t^1, \cdots, p_t^l$ to the language embedding space $\hat{p}_t^1, \cdots, \hat{p}_t^l$.

**Training Loss Function**    In summary, our training loss function is defined as follows:

$$\mathcal{L}(\theta, \phi) = \alpha_1 \mathcal{L}_{\text{language}}(\theta) + \alpha_2 \mathcal{L}_{\text{action}}(\theta) + \lambda \mathcal{L}_{\text{image}}(\theta, \phi), \tag{6}$$

where $\alpha_1, \alpha_2, \lambda$ are the weight hyperparameters of three components. In our experiments, we choose $\alpha_1 = 0.1$, $\alpha_2 = 10$, and $\lambda = 0.5$.

# 4 EXPERIMENTS

In this section, we validate through experiments on the autonomous driving simulation platform CARLA that VLA4CD can make fine-grained action decisions while maintaining dialogue functionality. We also examine the impact of each loss term in our loss function design on the performance of

*Other Sensors Input_0*: You can see that there is a car. It is speed, straight-line distance from you, and angle in the direction your heading are respectively  0.0 m/s,  12.4 m,  180.0 °.You are now -0.000m laterally away from your driving route.

*Question_0*:  What should be your next steering action?

*VLA4CD_pred_0*:  Your next steering action should be to turn the steering wheel to the left.

*Openvla_pred_0*:  your steering action, your steering action, your steering action, your steering action, your steering action, your steering action, your steering action, your steering action, your steering action, your steering action, your steering

*DriverGPT4_pred_0*:  000000000000000000000000000000000

*VLA4CD(no-image)_pred_0*:  right

*VLA4CD(no-language)_pred_0*:  -- ,- ,- ,- ,- ,- ,- ,- ,- ,- ,- ,- ,- ,-

_______________________________________________________________________

*Other Sensors Input_1*:  You can see that there are 2 cars. Their speed, straight-line distance from you, and angle in the direction your heading are respectively  5.07 2.4 m/s,  14.89 9.64 m,  14.18 179.72 °.You are now -0.029m laterally away from your driving route.

*Question_1*:  How does the detected car's speed impact your driving action?

*VLA4CD_pred_1*:  The detected car's speed impacts your driving action because if the detected car's speed is 0.000

*Openvla_pred_1*:  driving speed and your speed and your speed and your speed and your speed and your speed and

*DriverGPT4_pred_1*:  000000000000000000000000000000000

*VLA4CD(no-image)_pred_1*:  The car's speed does not impact my driving action.

*VLA4CD(no-language)_pred_1*:  -- ,- ,- ,- ,- ,- ,- ,- ,- ,- ,- ,- ,- ,-

Figure 2: An example of the VLA4CD question answering process

VLA4CD, as well as the quality of textual modality data in training data affects the decision-making performance of the model.

## 4.1 EXPERIMENTAL SETTING

We conducted our experiments in a benchmark environment called gym-carla (Chen, 2020), which is a third-party environment for OpenAI Gym, integrated with the closed-loop autonomous driving simulator CARLA 0.9.10 (Dosovitskiy et al., 2017). This experimental environment can provide image observations and supplementary textual descriptions relevant to the target task, with high demands on decision-making. During LoRA fine-tuning, we only fine-tuned the $Q$ projection and $V$ projection modules, the fine-tuned parameters accounting for only 0.06% of Llama-7B's whole parameters. For more details on the hyperparameter settings for VLA4CD, parameters for the linear mapping layer, and parameter settings in gym-carla, refer to Appendix A.1.

## 4.2 COMPARISON METHODS

The Behavior Cloning (BC) method performed in gym-carla (Chen, 2020) was used as a baseline. The other methods involved for comparison include RL methods Dreamer (Hafner et al., 2019) and Forbes (Chen et al., 2022), Decision Transformer (DT) (Chen et al., 2021), and VLA models OpenVLA (Kim et al., 2024) and DriverGPT4 (Xu et al., 2024).

## 4.3 TRAINING DATASETS

We trained all comparison methods based on an expert dataset $D_{expert}$, which is 5.69GB in size, containing 13,761 frames. We used 90% of it as the training set and the remaining as the test set. We evaluated these comparison methods online in the random mode of CARLA town03. Following the work on DT (Chen et al., 2021), we investigated the performance of sequence fusion for both single time steps and multiple time steps. We set the context length $H = 1$, resulting in a fusion sequence length of 489. This includes dividing the $128 \times 128$ image into 64 tokens and padding the text sequences to a length of 424 tokens, including an empty placeholder token. However, due to computational constraints, we only explored trajectory sequences with a maximum length of 489*4=1956 to validate performance in a longer context. We also explored whether the decision-making ability of VLA4CD is enhanced with longer context of trajectories in Appendix A.3. Additionally, we evaluated performance across different modalities and generalization capabilities in town04. For detailed information on the CARLA maps, refer to Appendix A.2. All comparison methods were tested online in the CARLA simulator. We conducted evaluations over 20 episodes, each consisting

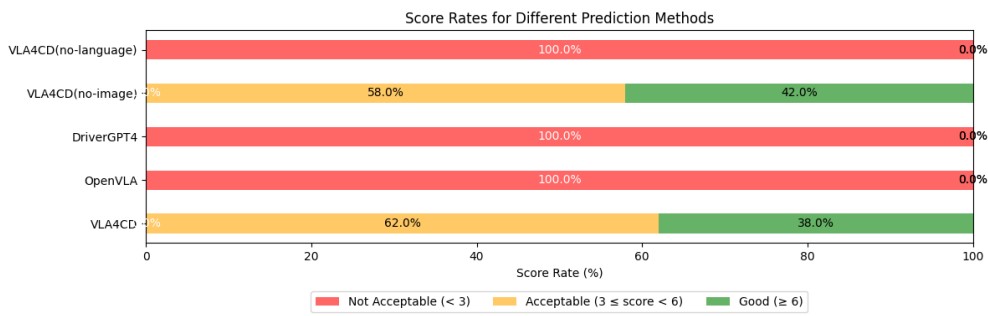

Figure 4: GPT-4o scores the answers from five methods for randomly generated inputs and question

of 1000 steps, with 200 involved vehicles, whose driving routes and met scenarios are generated in random mode.

The training dataset $D_{expert}$ was obtained from the EGADS framework (Tang et al., 2024), which designs RL and imitation learning-based agent with safety constraints, demonstrating excellent performance in CARLA. Therefore, we select this agent as our experts. We let such experts drive vehicles in town03 of CARLA to collect the dataset. Town03 is one complex map in CARLA, closely resembling real urban road environments, including various complex scenarios such as tunnels, intersections, roundabouts, curves, and multi-turns, covering an area of 400m × 400m, with a total road length of approximately 6km. As shown in Figure 3 (b), we used the layout of the town03 map for training. In the experimental environment for data collection and online evaluation, all vehicles randomly select directions at intersections, follow randomly generated routes, slow down for preceding vehicles, and stop when the traffic light ahead turns red.

Following Chen et al. (2024), we design a template based parser that translates sensor data (such as position and distance information, excluding vision and lidar) into natural language descriptions, as shown in *"other sensors input"* in Figures 1 and 2. For details on the templates, refer to Appendix A.6. Note that such *"other sensors input"* does not include any action-related information from VLA4CD, such as speed and heading angle. In this way, we can test whether VLA4CD can leverage informative text data to enhance the quality of action decisons.

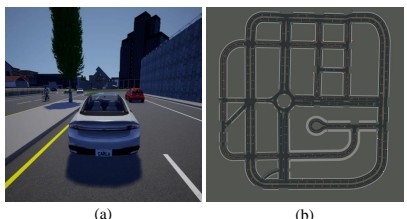

(a)       (b)

Figure 3: The (a) shows a sample view of the simulation environment, while the (b) presents a bird-eye view of our task scenario.

## 4.4 PERFORMANCE METRICS

**Performance metrics for evaluating the chatting ability** We used the powerful model GPT-4o (OpenAI, 2023) to compare the quality of answers given by VLA4CD with baseline models. Specifically, we first chose 50 pieces of randomly generated environment information and questions in CARLA. Then, given a piece of environment information and a question, we let VLA4CD and a baseline model each generate an answer. Then we used GPT-4o to score them, with a maximum score of 10. The scoring criteria are as follows: Not Acceptable ($< 3$), Acceptable ($3 \leq$ score $< 6$), Good ($\geq 6$). Additionally, to assess the impact of the language and image components on dialogue capabilities, we included VLA4CD (no-language) and VLA4CD (no-image), two simplified versions of VLA4CD trained by removing the loss items corresponding to text generation and image reconstruction, respectively: $\mathcal{L}_{\text{action}} + \mathcal{L}_{\text{image}}$, $\mathcal{L}_{\text{action}} + \mathcal{L}_{\text{language}}$.

**Performance metrics for evaluating the decision-making ability** We deployed our trained model on a car for use in navigating through a town. We considered commonly used metrics to evaluate the driving performance, including Collision Rate (CR), Off-road Rate (OR), Episode Completion Rate (ER), Average Safe Driving Distance (ASD), Average Reward (AR), and Driving Score (DS). DS is a composite indicator reflecting the overall performance of the vehicle in terms of safety, efficiency, and compliance with traffic rules. In addition, we use the reward function $f$ as described

Table 1: Evaluation results for different methods in town03 (random), *H=1*

| Method | Input | DS ↑ | AR (f) ↑ | ASD(m) ↑ | ER(%) ↑ | OR(%) ↓ | CR(%) ↓ |
|---|---|---|---|---|---|---|---|
| BC | image | 20.21±7.46 | 175.34±72.86 | 54.21±6.41 | 9.08±0.56 | 54.86±20.04 | 60.00±11.23 |
| DriverGPT4 | image, text | - | - | - | - | - | - |
| Openvla | image, text | -13.02±4.02 | -199.16±38.73 | 24.34±5.02 | 5.25±0.39 | 24.36±4.17 | 95.00±0.00 |
| VLA4CD | image, text | 92.78±23.75 | 466.80±91.66 | 71.77±9.40 | 16.35±1.56 | 15.33±4.36 | 55.00±11.41 |

Table 2: Evaluation results for different methods in town03 (random), *H=4*

| Method | Input | DS ↑ | AR (f) ↑ | ASD(m) ↑ | ER(%) ↑ | OR(%) ↓ | CR(%) ↓ |
|---|---|---|---|---|---|---|---|
| BC | image | 36.39±13.37 | 314.66±86.02 | 64.08±10.48 | 9.04±0.62 | 37.56±16.44 | 45.00±11.41 |
| Dreamer | image | -0.03±0.01 | -14.96±0.09 | 0.02±0.01 | 0.22±0.01 | 0.00±0.00 | 0.00±0.00 |
| Forbes | image | 0.98±1.43 | 21.63±21.72 | 22.84±1.00 | 6.30±0.31 | 18.78±1.03 | 56.67±9.20 |
| DT | image | 7.68±3.24 | 51.97±29.33 | 23.74±2.47 | 9.92±0.71 | 10.31±2.32 | 65.00±10.94 |
| DriverGPT4 | image, text | - | - | - | - | - | - |
| Openvla | image, text | -7.84±0.67 | -160.37±7.85 | 18.03±1.92 | 4.76±0.19 | 20.77±3.36 | 100.00±0.00 |
| VLA4CD | image, text | 105.25±14.03 | 349.52±49.75 | 59.76± 5.04 | 25.02±2.57 | 19.93±2.11 | 30.00±10.51 |

Table 3: Evaluation the generalization for different methods in town04 (random), *H=4*

| Method | Input | DS ↑ | AR (f) ↑ | ASD(m) ↑ | ER(%) ↑ | OR(%) ↓ | CR(%) ↓ |
|---|---|---|---|---|---|---|---|
| BC | image | 39.22±11.64 | 358.79±79.59 | 63.08±9.37 | 8.69±0.56 | 5.64±1.26 | 60.00±11.23 |
| Dreamer | image | -0.03±0.01 | -15.03±0.07 | 0.02±0.01 | 0.01±0.21 | 0.01±0.00 | 0.00±0.00 |
| Forbes | image | -2.63±2.75 | -17.37±22.98 | 19.79±1.20 | 6.24±0.69 | 15.80±2.74 | 66.70±8.75 |
| DT | image | 10.66±3.26 | 85.58±27.04 | 24.94±2.92 | 10.55±0.58 | 11.38±2.15 | 55.00±11.41 |
| DriverGPT4 | image, text | - | - | - | - | - | - |
| Openvla | image, text | -6.74±0.88 | -153.35±10.26 | 13.62±1.86 | 4.26±0.17 | 15.70±2.71 | 100.00±0.00 |
| VLA4CD | image, text | 94.26±15.26 | 384.52±51.72 | 56.93±4.03 | 21.49±1.86 | 12.75±2.28 | 45.00±11.41 |

in Chen et al. (2019) in the AR metric for training RL baselines. This reward function scores yaw, collisions, speeding, and lateral velocity for ego vehicle. Finally, we selected the checkpoint with the highest DS and AR score. For details, refer to Appendix A.4 and Appendix A.5.

## 4.5 EXPERIMENTAL RESULT ON CHATTING ABILITY EVALUATION

As shown in Figures 2 and 4, VLA4CD performs significantly better that others in terms of chatting ability. In contrast, OpenVLA performs poorly in question-answering because it focuses solely on optimizing the action loss. DriverGPT4 faces challenges as both tasks share the same decoder, causing the model to misinterpret inputs as only for action prediction, making it difficult to generate complete text. Despite having two independent loss items, the model has not effectively balanced these two losses. Furthermore, VLA4CD (no language) shows a significant gap in conversational ability compared to VLA4CD, while VLA4CD (no image) performs similarly to VLA4CD, highlighting the importance of the language loss component for enhancing chatting abilities.

## 4.6 EXPERIMENTAL RESULT ON DECISION-MAKING ABILITY EVALUATION

We define the "-" in Tables 1, 2, and 3 as a failure standard if a complete action value is not generated within 50 seconds. As shown in Table 1, VLA4CD significantly outperforms BC and OpenVLA in terms of DS, AR, and ASD at a single time step, while DriverGPT4 fails to generate precise action values. VLA4CD also shows significant improvements over other methods across multiple time steps in Table 2, indicating sustained benefits over longer durations. We evaluated these models' generalization capability by training them on the town03 dataset and then evaluating them online in town04. As shown in Table 3, the primary metric DS of VLA4CD significantly exceeds that of the other methods, showcasing its strong generalization ability. Tables 1, 2, and 3 indicate that DriverGPT4 faces challenges in generating precise action values for real-time control commands, highlighting the difficulties of directly generating accurate values using a detokenizer. In contrast, OpenVLA can generate precise values in experiments but produces identical action commands, causing vehicles to wander or spin in a place, resulting in significant penalties. Results in Tables 2

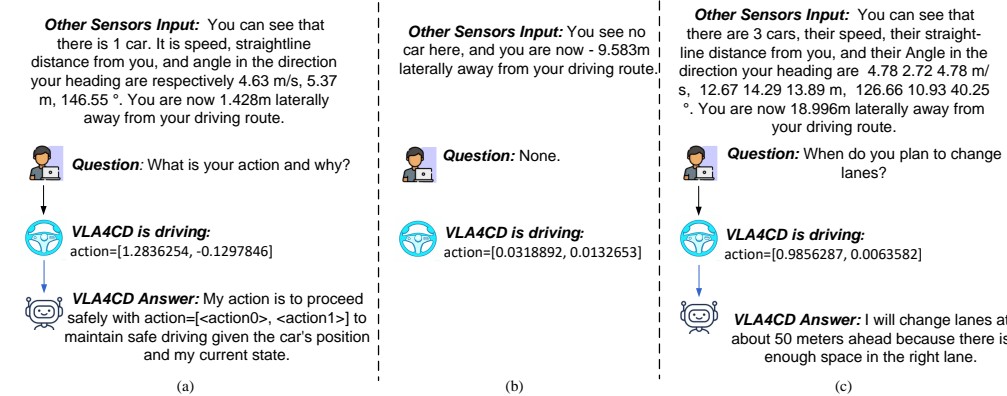

Figure 5: An example show on how VLA4CD smoothly engages in conversation with a human while simultaneously making real-time action decisions during the driving process

Table 4: Ablation studies on the loss function of VLA4CD in town03 (random), *H=4*

| Loss function | Input | DS ↑ | AR ($f$) ↑ | ASD(m) ↑ | ER(%) ↑ | OR(%) ↓ | CR(%) ↓ |
|---|---|---|---|---|---|---|---|
| $\mathcal{L}_{\text{image}}+\mathcal{L}_{\text{language}}+\mathcal{L}_{\text{action-bins}}$ | image, text | 11.57±0.00 | 142.83±0.01 | 22.71± 0.01 | 8.10±0.05 | 30.87±0.10 | 100.00±0.00 |
| $\mathcal{L}_{\text{image}}+\mathcal{L}_{\text{action}}$ | image, text | 45.08±10.88 | 234.36±52.21 | 39.64± 4.03 | 14.13±1.71 | 16.68±3.15 | 30.00±10.51 |
| $\mathcal{L}_{\text{language}}+\mathcal{L}_{\text{action}}$ | image, text | 74.85±10.97 | 331.78±49.88 | 50.63±4.73 | 18.62±1.95 | 15.96±2.45 | 25.00±9.93 |
| $\mathcal{L}_{\text{image}}+\mathcal{L}_{\text{language}}+\mathcal{L}_{\text{action}}(our)$ | image, text | 105.25±14.03 | 349.52±49.75 | 59.76± 5.04 | 25.02±2.57 | 19.93±2.11 | 30.00±10.51 |

and 3 demonstrated that VLA4CD also significantly outperforms DT, Dreamer, and Forbes in terms of decision-making and model generalization.

Finally, Figure 5 illustrates how our model smoothly engages in conversation with a human while simultaneously making real-time action decisions during the driving process.

## 4.7 ABLATION STUDIES ON THE LOSS FUNCTION DESIGN

As shown in Equation (6), our loss function is composed of three losses, namely action loss $\mathcal{L}_{\text{action}}$, language loss $\mathcal{L}_{\text{language}}$, and image loss $\mathcal{L}_{\text{image}}$. We conducted ablation studies to investigate the effect of each loss on the performance of VLA4CD. The experiment result is shown in Table 4, where the action-bins loss $\mathcal{L}_{\text{action-bins}}$ denotes the action loss used by OpenVLA and RT2. They deal with continuous valued actions by value discretization. We included VLA4CD (no-language) and VLA4CD (no-image), two simplified versions of VLA4CD trained by using $\mathcal{L}_{\text{action}} + \mathcal{L}_{\text{image}}$ and $\mathcal{L}_{\text{action}} + \mathcal{L}_{\text{language}}$, respectively.

**On the effect of $\mathcal{L}_{\text{action}}$**   As shown in Table 4, if we compare the performance metrics of $\mathcal{L}_{\text{image}} + \mathcal{L}_{\text{language}} + \mathcal{L}_{\text{action-bins}}$ with that of $\mathcal{L}_{\text{image}} + \mathcal{L}_{\text{language}} + \mathcal{L}_{\text{action}}$, we can see a clear advantage of using our action loss $\mathcal{L}_{\text{action}}$ over using $\mathcal{L}_{\text{action-bins}}$. This explains why VLA4CD outperforms VLA models that use the type of action loss similar to $\mathcal{L}_{\text{action-bins}}$, as shown in Tables 1, 2, and 3. Specifically, from our experiments, we found that doing action discretization and tokenization as in current VLA models lead to low training loss but bad inference performance. This is because adjacent action intervals are represented by consecutive token IDs (e.g., 31830 and 31831), which are close in token space. Consequently, the model tends to output the same token (31830 or 31831) in inference, while the actual action values corresponding to them can have significant differences. In contrast, our approach proposed to deal with continuously valued actions can avoid this phenomenon to happen.

**On the effect of $\mathcal{L}_{\text{language}}$**   As shown in Table 4, if we compare performance metrics between $\mathcal{L}_{\text{image}} + \mathcal{L}_{\text{action}}$ (corresponding to VLA4CD (no-language)) and $\mathcal{L}_{\text{image}} + \mathcal{L}_{\text{language}} + \mathcal{L}_{\text{action}}$ (corresponding to VLA4CD), we see that including $\mathcal{L}_{\text{language}}$ in the loss function significantly enhances the quality of decision-making. As shown in Figures 2 and 4, VLA4CD (no-language) has significantly different dialogue capabilities compared to VLA4CD, while VLA4CD (no-image) performs similarly to VLA4CD. It demonstrates that $\mathcal{L}_{\text{language}}$ plays an important role for maintaining the dialogue

Table 5: The impact of the quality of textual modality data in training data on the decision-making performance of VLA4CD

| Input | Noise ratio | DS ↑ | AR ($f$) ↑ | ASD(m) ↑ | ER(%) ↑ | OR(%) ↓ | CR(%) ↓ |
|---|---|---|---|---|---|---|---|
| image, text | 100% | -0.01±1.12 | -5.10±0.00 | 0.00±0.00 | 0.30±0.00 | 0.00±0.00 | 0.00±0.00 |
| image, text | 75% | 2.74±2.17 | 16.93±29.84 | 18.63±1.70 | 7.38±0.32 | 16.93±2.32 | 55.0±11.41 |
| image, text | 50% | 4.41±1.87 | 49.12±12.56 | 6.10±1.34 | 6.35±0.67 | 0.00±0.00 | 0.00±0.00 |
| image, text | 25% | 15.58±2.49 | 143.70±23.54 | 23.23±3.24 | 8.25±1.11 | 26.75±1.83 | 10.00±6.88 |
| image, text | 0% | 93.89±29.73 | 336.11±86.72 | 45.42±9.53 | 16.68±2.50 | 19.05±4.96 | 5.00±5.00 |

capability. To summarize, including $\mathcal{L}_{\text{language}}$ in the loss function has beneficial impacts on both dialogue and decision-making.

**On the effect of $\mathcal{L}_{\text{image}}$** As shown in Table 4, when we added $\mathcal{L}_{\text{image}}$ in the loss function (corresponding to results of $\mathcal{L}_{\text{image}} + \mathcal{L}_{\text{language}} + \mathcal{L}_{\text{action}}$), all performance metrics related to decision-making are increased in value, compared to $\mathcal{L}_{\text{language}} + \mathcal{L}_{\text{action}}$. This confirms that the $\mathcal{L}_{\text{image}}$ indeed brings remarkable benefits for enhancing decision-making performance. We argue that this is because, during the decision-making, doing high-quality image reconstruction can further explore and utilize the rich information related to the current scene within the image modality data, thereby benefiting the decision-making.

### 4.8 How the quality of textual modality data in training data affects the decision-making performance of the model?

Imagine a driver is operating a car, with a friend sitting inside the vehicle and conversing with the driver. If this friend provides valuable reminders, such as alerting the driver to a car approaching from the blind spot, the friend's words would be beneficial to the driver's decision-making. On the contrary, if the friend's words are irrelevant noise to the current situation, it might interfere with the driver's ability to make accurate decisions. Therefore, we designed a set of experiments to test whether our model exhibits similar performance to that of human drivers in decision making. The result is presented in Table 5. As is shown, when we add more and more noisy information unrelated to driving scenarios into the text modality data in the training dataset, the quality of the decisions output by our model rapidly decreases. This indicates that the performance of our model is very similar to that of human drivers.

## 5 Conclusion

In this paper, we investigated how to develop a multimodal pre-trained model that simultaneously achieves the dialogue function of LLM and the decision-making function of VLA. We use the autonomous driving scenario as an example to explain our problem setup and model development process. Unlike the instruction-following setup used behind VLA models, our problem setup can be described as making decisions while conversing. In the former, text data appears in the form of instructions before the decision-making process; in the latter, text data and decision data are interwoven (imagine a large pre-trained model making driving decisions while chatting with people in the car). For the aforementioned problem setup, we provide a method for constructing a multimodal Visual Language Action model for simultaneously Chatting and Decision making (VLA4CD). Experimental results show that, thanks to our proposed way to deal with continuous valued actions, our design of the training cost function, and the use of label smoothing technique, our VLA4CD model significantly outperforms the SOTA VLA model, RL, and decision transformer methods in decision-making performance, while also possessing smooth dialogue capabilities.

VLA4CD can be seen as a functional extension of the VLA model, while its performance depends on the quality of the training data set. Interesting future research directions include: further testing and validation using large-scale real-world driving datasets; and applying our approach to scenarios beyond autonomous driving, such as home robots.

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

# A APPENDIX

## A.1 HYPERPARAMETER SETTINGS

In this section, we respectively introduce the model parameters of VLM4EDM, the parameters of the custom linear layers, as well as the parameters of gym-carla and evaluation, as shown in Tables 6, 7, and 8.

Table 6: Hyperparameters

| Parameter | Value |
|---|---|
| batch_size | 64 |
| micro_batch_size | 8 |
| num_epochs | 3 |
| learning_rate | 3e-4 |
| cutoff_len | 424 |
| val_set_size | 0.1 |
| save_step | 25 |
| lora_r | 8 |
| lora_alpha | 16 |
| lora_dropout | 0.05 |
| lora_target_modules | {q_proj, k_proj} |
| Other Sensors Input_types | {obs, text} |
| lambda_action | 10 |
| lambda_smooth | 0.1 |
| lambda_img | 0.5 |
| horizon | 1 |
| regular_action_loss | False |
| img_patch_size | 16 |

Table 7: Model Parameters and Layers

| Parameter/Layer | Details |
|---|---|
| num_patches | 64 |
| tokenizer_vocab_size | 32000 |
| split_obs_proj | Conv2d(3, 4096, kernel_size=16, stride=16) |
| inverse_split_obs_proj | ConvTranspose2d(4096, 3, kernel_size=16, stride=16) |
| split_obs_position_embedding | Parameter(torch.randn(1, 64, 4096)) |
| text_embedding | nn.Embedding(32000, 4096) |
| custom_lm_head | Linear(4096, 32000, bias=False) |
| actor_linear1 | Linear(4096, 2048) |
| actor_linear2 | Linear(2048, 1024) |
| actor_linear3 | Linear(1024, 512) |
| actor_linear4 | Linear(512, 256) |
| actor_linear5 | Linear(256, 128) |
| actor_linear6 | Linear(128, 64) |
| actor_linear7 | Linear(64, 2) |
| reconstruction_layer | Linear(4096, micro_batch_size*3*128*128) |
| action_linear | Linear(2, 4096) |

## A.2 CARLA MAPS

In order to comprehensively evaluate the performance of our EGADS, we utilized five maps in CARLA, including town03, town04 as shown in Figure 6. Town03 is a larger town with features of a downtown urban area. The map includes some interesting road network features such as a

Table 8: gym-carla and evaluation Environment Parameters

| Parameter | Value |
|---|---|
| Number of Vehicles | 200 |
| Number of Walkers | 0 |
| Random Seed | 1 |
| Other Sensors Input_names | lidar_noground |
| Display Size | 400 |
| Max Past Step | 1 |
| Time Step (dt) | 0.1 |
| Discrete Control | False |
| Continuous Acceleration Range | [-3.0, 3.0] |
| Continuous Steering Range | [-0.2, 0.2] |
| Ego Vehicle Filter | vehicle.lincoln* |
| Traffic Manager Port | Random integer (2000 to 9000) |
| Town Map | town03 or town04 |
| Task Mode | Random |
| Max Time per Episode | 2000 |
| Max Waypoints | 12 |
| Observation Range | 32 |
| LiDAR Bin Size | 0.25 |
| Distance Behind Ego Vehicle | 12 |
| Lane Threshold | 2.0 |
| Desired Speed | 8 |
| Max Ego Vehicle Spawn Times | 200 |
| Display Route | True |
| PIXOR Grid Size | 64 |
| PIXOR Mode | False |
| Predict Speed | True |

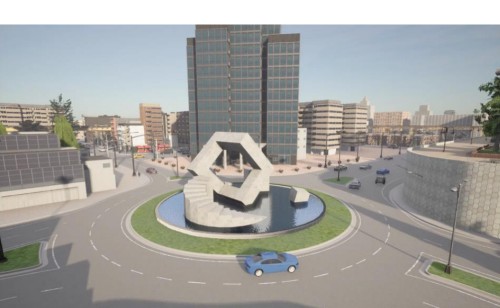 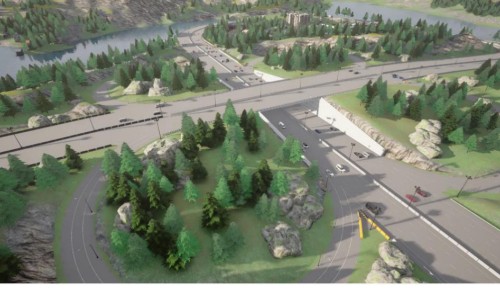

(a) Town03          (b) Town04

Figure 6: CARLA maps

roundabout, underpasses and overpasses. The town also includes a raised metro track and a large building under construction. Town04 is a small town with a backdrop of snow-capped mountains and conifers. A multi-lane road circumnavigates the town in a "figure of 8".

### A.3 IS MODEL DECISION-MAKING ABILITY ENHANCED WITH LONGER CONTEXT OF TRAJECTORIES?

As shown in Table 9, we observed that although the context length $H$ of input trajectories is longer, the overall DS and AR of VLA4CD show some improvement, but the increase is not significant. This improvement is primarily attributed to the higher route completion and lower collision rates associated with longer time steps. According to Section 4.3, when $H = 4$, the sequence length

Table 9: Evaluation VLA4CD longer context results for mulitmodal input in town03 (random)

| Input | $\mathcal{L}_{\text{image}}$ | $H$ | DS ↑ | AR ($f$) ↑ | ASD(m) ↑ | ER(%) ↑ | OR(%) ↓ | CR(%) ↓ |
|---|---|---|---|---|---|---|---|---|
| image | × | 1 | 29.55±6.17 | 226.91±42.24 | 54.24±4.30 | 11.85±0.68 | 20.22±5.57 | 70.00±10.5 |
| image | × | 4 | 22.38±4.96 | 155.79±31.87 | 32.45±1.74 | 14.41±0.59 | 15.93±2.65 | 40.00±11.23 |
| text | × | 1 | 37.44±10.11 | 248.89±52.91 | 47.37±5.43 | 15.63±1.98 | 17.02±2.71 | 40.00±11.24 |
| text | × | 4 | 44.16±7.39 | 252.10±38.94 | 46.96±3.23 | 15.66±1.06 | 12.86±2.45 | 60.00±11.23 |
| image, text | × | 1 | 68.10±13.20 | 417.24±57.41 | 58.81±6.55 | 13.71±1.26 | 11.39±2.41 | 40.00±11.24 |
| image, text | × | 4 | 74.85±10.97 | 331.78±49.88 | 50.63±4.73 | 18.62±1.95 | 15.96±2.45 | 25.00±9.93 |
| image, text | ✓ | 1 | 92.78±23.75 | 466.80±91.66 | 71.77±9.40 | 16.35±1.56 | 15.33±4.36 | 55.00±11.41 |
| image, text | ✓ | 4 | 105.25±14.03 | 349.52±49.75 | 59.76± 5.04 | 25.02±2.57 | 19.93±2.11 | 30.00±10.51 |

extends to 1956, representing a fourfold increase in sequence length. Despite this, the improvement in DS and AR scores is not pronounced. Notably, in metrics such as AR and ADS, the performance of $H = 4$ is even worse than that of $H = 1$. This suggests that the input information might be redundant, and excessively long trajectories could negatively impact decision-making ability.

This result highlights several key issues. First, while longer context lengths provide the model with more historical context and information, an excessive amount of information may hinder the ability of model to effectively filter and extract useful decision signals, leading to information redundancy. Redundant information not only increases the computational complexity but also may distract the attention of model, reducing its capacity to capture critical features and thereby affecting overall decision-making. Therefore, shorter context length sequences provide more concise and precise inputs, facilitating quicker and more accurate judgments by the model. This indicates that the current fusion method has limited performance improvements. Chen et al. (2021); Hung et al. (2019) suggest that longer context lengths can bring more benefits for decision control, so we also consider how to compress historical information and efficiently fuse it in the future to enhance decision-making.

## A.4 REWARD FUNCTION

We use the default reward function of the Gym-Carla benchmark (Chen, 2020) to evaluate all experimental methods, as follows:

$$f = 200r_c + v_{lon} + 10r_f + r_o - 5\alpha^2 + 0.2r_{lat} - 0.1 \tag{7}$$

where $r_c$ is the reward related to collision, which is set to -1 if the ego vehicle collides and 0 otherwise. $v_{lon}$ is the longitudinal speed of the ego vehicle. $r_f$ is the reward related to running too fast, which is set to $-1$ if it exceeds the desired speed (8 m/s here) and 0 otherwise. $r_o$ is set to $-1$ if the ego vehicle runs out of the lane, and 0 otherwise. $\alpha$ is the steering angle of the ego vehicle in radians. $r_{lat}$ is the reward related to lateral acceleration, which is calculated by $r_{lat} = -|\alpha| \cdot v_{lon}^2$. The last constant term is added to prevent the ego vehicle from standing still.

## A.5 MEASURE PERFORMANCE METRICS

We use multiple key metrics to evaluate the performance of autonomous driving models in various driving scenarios. Collision Rate (CR): the frequency at which the vehicle collides with obstacles or other vehicles. This metric is critical for assessing the safety of the driving model. Outlane Rate (OR): the rate at which the vehicle deviates from its designated lane. This metric evaluates the ability of modes to maintain proper lane discipline. Episode Completion Rate (ER): the percentage of driving tasks or episodes that the vehicle successfully completes. Higher completion rates indicate better task performance. Average Safe Driving Distance (ASD): the average distance driven without incidents, such as collisions or off-road events. This metric highlights the capability to drive safely over extended periods. Average Return (AR): A metric that measures the cumulative reward collected by the vehicle during its driving tasks, often reflecting both task performance and adherence to safety guidelines. Driving Score (DS): A comprehensive metric that reflects the overall performance of the vehicle in terms of safety, efficiency, and compliance with traffic rules.

$$CR = \frac{N_{\text{collisions}}}{N_{\text{total\_episodes}}}, OR = \frac{N_{\text{off\_road\_events}}}{N_{\text{total\_episodes}}}, ER = \frac{N_{\text{completed\_steps}}}{N_{\text{total\_steps}}} \tag{8}$$

$$ASD = \frac{\sum_{i=1}^{N_{\text{episodes}}} \text{distance}_i}{N_{\text{total\_episodes}}}, DS = ER \times AR \tag{9}$$

Where $N_{\text{collisions}}$ is the number of collisions during the episode, and $N_{\text{total\_episodes}}$ is the total number of episodes in the test. Where $N_{\text{off\_road\_events}}$ is the number of times the vehicle went off-road, and $N_{\text{total\_steps}}$ is the total number of episodes. Where $\text{distance}_i$ is the distance driven during the $i$-th safe driving episode, and $N_{\text{safe\_episodes}}$ is the number of episodes without incidents (such as collisions or off-road events). Where $N_{\text{completed\_steps}}$ is the number of successfully completed steps, and $N_{\text{total\_steps}}$ is the total number of steps in the episode. Where $AR$ is the average reward $f$ collected during the episode.

## A.6 The natural language template for text input

We obtained information from the CARLA environment using other sensors (such as speed sensors and position sensors), excluding the acceleration and steering (action) of the ego vehicle). This information is transformed into a natural language template that the VLA can understand, as shown below:

*<lateral_dis, delta_yaw, speed, vehicles_info> = <observation_vehicle_state>*
*<vehicles_num> = <len(vehicles_info)>*
*<multi_dis += str(vehicles_info[i][0])+"", multi_yaw += str(vehicles_info[i][1])+"", multi_speed += str(vehicles_info[i][2])+"">*
*<**if vehicles_num**=1:>*
*<**new_input**="You can see that there is a car. It is speed, straight-line distance from you, and angle in the direction your heading are respectively {**multi_speed**} m/s, {**multi_dis**} m, {**multi_yaw**}°." "You are now {**lateral_dis**}m laterally away from your driving route. ">*
*<**elif vehicles_num**>1:>*
*<**new_input**="You can see that there are **vehicles_num** cars. Their speed, straight-line distance from you, and angle in the direction your heading are respectively {**multi_speed**} m/s, {**multi_dis**} m, {**multi_yaw**}°." "You are now {**lateral_dis**}m laterally away from your driving route. ">*
*<**elif vehicles_num**=0:>*
*<**new_input**="You see no car here, and you are now {**lateral_dis**}m laterally away from your driving route.">*

## A.7 The benefits of cross-entropy loss and label smoothing loss for VLA4CD

We found that merely replacing specific numerical values in the translation template (Chen et al., 2024) results in minimal representational differences caused by the sequential nature of data, making it easy for conventional cross-entropy loss to lead to overfitting in text generation tasks. As shown in Table 10, we tested on both town03 and town04, which led to a decline in the decision-making performance of model. Compared to cross-entropy loss, cross-entropy loss with smoothed labels performed better. Therefore, we chose cross-entropy loss with smoothed labels as the loss for text generation in VLA4CD in our experiments.

## A.8 The impact of training data-related factors on the decision performance of model

In the multimodal ablation experiments on the VLA4CD model, as shown in Table 11, we systematically removed or replaced individual modalities to evaluate their contribution to decision-making. The results show that models utilizing image and text fusion significantly outperform those with only a single image or text input in terms of decision accuracy and stability. This indicates that the text modality in our dataset provides higher-level semantic abstraction to complement visual inputs, thereby enhancing overall decision-making ability. In addition, as shown in Table 11, a single text input performs better than a single image input, indicating that the information provided by the text modality in our dataset (especially from *"other sensors input"*, as shown in Figure 2) is highly beneficial for improving the decision-making ability of model.

Table 10: We evaluated the performance of VLA4CD using smooth label loss and cross-entropy loss functions, *H=4*

| $\mathcal{L}_{\text{language}}$ | Town | DS ↑ | AR ($f$) ↑ | ASD(m) ↑ | ER(%) ↑ | OR(%) ↓ | CR(%) ↓ |
|---|---|---|---|---|---|---|---|
| Cross Entropy | town03 | 48.97±7.60 | 296.53±40.72 | 47.10±4.87 | 15.37±0.85 | 12.41±2.73 | 35.00±10.94 |
| Smooth Label | town03 | 105.25±14.03 | 349.52±49.75 | 59.76± 5.04 | 25.02±2.57 | 19.93±2.11 | 30.00±10.51 |
| Cross Entropy | town04 | 66.69±16.97 | 358.11±61.10 | 52.72±5.44 | 15.43±1.11 | 9.63±1.42 | 55.00±11.41 |
| Smooth Label | town04 | 94.26±15.26 | 384.52±51.72 | 56.93±4.03 | 21.49±1.86 | 12.75±2.28 | 45.00±11.41 |

Table 11: Evaluating the impact of different modal inputs on the decision-making of VLA4CD in town03 (random), *H=4*

| Input | $\mathcal{L}_{\text{image}}$ | DS ↑ | AR ($f$) ↑ | ASD(m) ↑ | ER(%) ↑ | OR(%) ↓ | CR(%) ↓ |
|---|---|---|---|---|---|---|---|
| image | × | 22.38±4.96 | 155.79±31.87 | 32.45±1.74 | 14.41±0.59 | 15.93±2.65 | 40.00±11.23 |
| text | × | 44.16±7.39 | 252.10±38.94 | 46.96±3.23 | 15.66±1.06 | 12.86±2.45 | 60.00±11.23 |
| image, text | × | 74.85±10.97 | 331.78±49.88 | 50.63±4.73 | 18.62±1.95 | 15.96±2.45 | 25.00±9.93 |
| image, text | ✓ | 105.25±14.03 | 349.52±49.75 | 59.76± 5.04 | 25.02±2.57 | 19.93±2.11 | 30.00±10.51 |

A.9    THE NOISE CONSISTED OF INFORMATION DATASETS

The noise consisted of information completely unrelated to the current driving scenario as follow: {"A playful puppy brings joy and laughter to our days", "The whisper of the wind carries secrets of the universe", "A hidden garden blooms with the magic of nature's colors", "The aroma of fresh coffee awakens the senses each morning", "A handwritten letter feels like a warm hug from afar", "The glimmer of fireflies creates a magical summer night", "A spontaneous adventure can lead to unforgettable memories", "The serenity of a quiet lake reflects the beauty of the world", "A gentle touch can convey love without a single word", "The laughter of friends is the sweetest melody of all", "A warm hug is a universal language of comfort", "The dance of leaves in the breeze tells stories of change", "A cozy fire invites stories and shared moments", "The beauty of art inspires creativity and self-expression", "A day spent volunteering fills the heart with purpose", "The excitement of a new book is like embarking on a journey", "A delicious meal shared brings people closer together", "The sound of laughter can brighten even the gloomiest day", "A fleeting moment can hold the weight of a thousand memories", "The charm of small towns lies in their simple beauty", "A gentle rain nurtures the earth and inspires growth", "A colorful painting captures the essence of joy", "The peace of a mountain retreat refreshes the soul", "A favorite mug holds warmth and comfort on a chilly day", "The rustle of leaves underfoot reminds us of nature's rhythm", "A well-crafted story has the power to transport us anywhere", "The thrill of discovery keeps our spirits young and curious", "A cherished photograph holds a lifetime of memories", "The beauty of winter blankets the world in quiet calm", "A moment of kindness can change the trajectory of a day", "The aroma of spices fills the kitchen with warmth and love", "A shared joke creates bonds that laughter alone cannot", "The glow of a sunrise fills the heart with hope", "A melody can linger in the mind long after it fades", "The colors of autumn leaves create a vibrant tapestry", "A soft pillow cradles the head and invites sweet dreams", "The laughter of children brings joy and light to our lives", "A surprise visit from a friend can brighten any day", "The beauty of a flower garden is a celebration of life", "A good book can be a loyal companion on lonely nights", "The embrace of nature can heal and rejuvenate the spirit", "A treasure hunt ignites the spirit of adventure", "The warmth of homemade cookies fills the home with love", "A playful kitten brings joy and mischief to our lives", "The scent of pine trees evokes memories of the forest", ]}

A.10    OUR DEFINED SET OF PROBLEMS

Randomly selected a question from the set of questions.  = {"What are you seeing/observing?", "What are you paying attention to and why?", "Are there any traffic lights? What's the color of the traffic light?", "What's your current speed and steering angle?", "What is your action and why?", "Summarize the current driving scenario at a high level.", "How are you going to drive in this situation and why?", "What's the straight-line distance to the nearest car?", "What is the angle of the nearest car relative to your heading?", "Is there any lateral deviation from your driving route?", "What should be your next steering action?", "What should be your next acceleration command?", "Is there any moving object around you?", "Describe the position of the car relative to your head-

ing.", "What is your current lateral position relative to your route?", "What would be a safe driving action given the detected car's details?", "What is the speed of the detected car?", "How far is the detected car from you?", "What angle should you adjust your steering to avoid collision?", "Why is it important to note the angle of the detected car?", "Is the detected car in motion?", "What should you be cautious of given the car's position?", "What action should be taken to maintain alignment on your driving route?", What should you avoid in this situation to prevent collision?, "What considerations are necessary for the detected car's speed?", "What's the importance of your current lateral position in planning the next action?", Why did you brake just now?, "What factors are influencing your next driving decision?", "Is there any obstacle directly ahead?", "How should you interpret the car's angle for your steering decision?", "What immediate adjustments are necessary for safe driving?", "How does the detected car's speed impact your driving action?", "What should be your focus given the detected car's proximity and angle?", "What safe action is suggested based on the current scenario?", "What should you avoid in this situation to prevent collision?", "Is there a need for a speed adjustment?", "How will your steering angle change based on the detected car's angle?", "What should you consider for maintaining a safe path?", "How would you describe the current traffic conditions?", "What immediate action is necessary given your current lateral position?", "What factors need to be monitored to ensure safe navigation?", "Is the detected car influencing your path directly?", "What is the priority in adjusting your speed and direction?", "    "}

## A.11 FURTHER RESULTS ON VLA4CD IN ANSWERING QUESTIONS

Other Sensors Input_32: You can see that there is a car. It is speed, straight-line distance from you, and angle in the direction your heading are respectively 2.38 m/s, 9.77 m, 179.84 °.You are now -0.008m laterally away from your driving route.
Question_32: What would be a safe driving action given the detected car's details?
VLA4CD_pred_32: The car is speed, straight-line distance from you
Openvla_pred_32: detected car's from the detected car's and the detected car's from the detected car's
DriverGPT4_pred_32: 0000000000000000000000000000000000
VLA4CD_image_pred_32: The car is speed, straight-line distance from you, and angle in the direction your heading are respectively 2.38 m/s, 9.77 m, 179.84 °.You are now -0.008m laterally away from your driving route.
VLA4CD_language_pred_32: - -

---

Other Sensors Input_33: You can see that there is a car. It is speed, straight-line distance from you, and angle in the direction your heading are respectively 2.45 m/s, 9.73 m, 179.82 °.You are now -0.009m laterally away from your driving route.
Question_33: What factors need to be monitored to ensure safe navigation?
VLA4CD_pred_33: You need to monitor the speed, straight-line distance from you, and angle in the direction your heading are respectively 2.45 m/s, 9.73 m, 179.82 °.
Openvla_pred_33: monitored to safe to ensure safe and to ensure safe monitored to safe
DriverGPT4_pred_33: 0000000000000000000000000000000000
VLA4CD_image_pred_33: speed, straight-line distance from you, and angle in the direction your heading are respectively 2.45 m/s, 9.73 m, 179.82 °.You are now -0.009m laterally away from your driving route.
VLA4CD_language_pred_33: -

---

Other Sensors Input_34: You can see that there is a car. It is speed, straight-line distance from you, and angle in the direction your heading are respectively 2.49 m/s, 9.69 m, 179.81 °.You are now -0.011m laterally away from your driving route.
Question_34: What should be your next steering action?
VLA4CD_pred_34: Your next steering action should be to turn the steering wheel to the left.
Openvla_pred_34: your steering action and your next steering action and your steering action and your steering action and your steering action and
DriverGPT4_pred_34: 000000000000000000000000000000000000000000000000
VLA4CD_image_pred_34: right

VLA4CD_language_pred_34: -

---

Other Sensors Input_35: You can see that there is a car. It is speed, straight-line distance from you, and angle in the direction your heading are respectively 2.47 m/s, 9.66 m, 179.79 °.You are now -0.012m laterally away from your driving route.
Question_35: What's the importance of your current lateral position in planning the next action?
VLA4CD_pred_35: The current lateral position is important in planning the next action because if you are too far away from your driving route.
Openvla_pred_35: your current position in planning the next action
DriverGPT4_pred_35: 000000000000000000000000000000000
VLA4CD_image_pred_35: Your current lateral position is important in planning the next action because if you don't plan the next action, you will not be able to reach your destination.
VLA4CD_language_pred_35:

---

Other Sensors Input_36: You can see that there is a car. It is speed, straight-line distance from you, and angle in the direction your heading are respectively 2.41 m/s, 9.64 m, 179.78 °.You are now -0.014m laterally away from your driving route.
Question_36: What safe action is suggested based on the current scenario?
VLA4CD_pred_36: You can see that there is a car. It is speed, straight-line distance from you, and angle in the direction your heading are respectively.
Openvla_pred_36: suggested based on the current safe action on the current
DriverGPT4_pred_36: 000000000000000000000000000000000
VLA4CD_image_pred_36: The car is speed, straight-line distance from you, and angle in the direction your heading are respectively 2.41 m/s, 9.64 m, 179.78 °.You are now -0.014m laterally away from your driving route.
VLA4CD_language_pred_36: - -

---

Other Sensors Input_37: You can see that there is a car. It is speed, straight-line distance from you, and angle in the direction your heading are respectively 2.37 m/s, 9.63 m, 179.77 °.You are now -0.016m laterally away from your driving route.
Question_37: Is there any lateral deviation from your driving route?
VLA4CD_pred_37: There is no lateral deviation from your driving route.
Openvla_pred_37: driving route from your route and your route from your driving route from your route from your
DriverGPT4_pred_37: 000000000000000000000000000000000
VLA4CD_image_pred_37: Yes, there is lateral deviation from your driving route.
VLA4CD_language_pred_37: -

---

Input_38: You can see that there is a car. It is speed, straight-line distance from you, and angle in the direction your heading are respectively 2.37 m/s, 9.63 m, 179.76 °.You are now -0.018m laterally away from your driving route.
Question_38: How will your steering angle change based on the detected car's angle?
VLA4CD_pred_38: Your steering angle will change based on the detected car's angle.
Openvla_pred_38: your steering on the detected car's steering
DriverGPT4_pred_38: 0000000000000000000000000000000000000000
VLA4CD_image_pred_38: The steering angle will change based on the detected car's angle.

---

Input_39: You can see that there is a car. It is speed, straight-line distance from you, and angle in the direction your heading are respectively 2.41 m/s, 9.63 m, 179.75 °.You are now -0.020m laterally away from your driving route.
Question_39: Is there any moving object around you?
VLA4CD_pred_39: Yes, there is a car. It is speed, straight-line distance from you, and angle in the direction your heading are respectively 2.
Openvla_pred_39: moving object of you and your work with your work?
DriverGPT4_pred_39: 0000000000000000000000000000000000000000000
VLA4CD_image_pred_39: Yes, there is a car.

VLA4CD_language_pred_39: - -

---

Input_40: You can see that there is a car. It is speed, straight-line distance from you, and angle in the direction your heading are respectively 2.45 m/s, 9.63 m, 179.74 °.You are now -0.022m laterally away from your driving route.
Question_40: Is there any moving object around you?
VLA4CD_pred_40: Yes, there is a car. It is speed, straight-line distance from you, and angle in the direction your heading are respectively 2.
Openvla_pred_40: moving object of you and your work with your work?
DriverGPT4_pred_40: 000000000000000000000000000000000000000000000
VLA4CD_image_pred_40: Yes, there is a car.
VLA4CD_language_pred_40: - -

---

Input_41: You can see that there is a car. It is speed, straight-line distance from you, and angle in the direction your heading are respectively 2.46 m/s, 9.63 m, 179.73 °.You are now -0.024m laterally away from your driving route.
Question_41: How does the detected car's speed impact your driving action?
VLA4CD_pred_41: The detected car's speed impacts your driving action because if the detected car's speed is faster than your driving speed
Openvla_pred_41: speed with your speed and your speed with your speed speed with your speed with your speed with your speed with your speed with
DriverGPT4_pred_41: 000000000000000000000000000000000000000000000
VLA4CD_image_pred_41: The car's speed impacts my driving action because the car's speed is increasing, which means that the car's speed is getting faster
VLA4CD_language_pred_41: - -

---

Input_42: You can see that there is a car. It is speed, straight-line distance from you, and angle in the direction your heading are respectively 2.43 m/s, 9.63 m, 179.73 °.You are now -0.027m laterally away from your driving route.
Question_42: What is your action and why?
VLA4CD_pred_42: My action is to turn left. I am turning left because I want to turn left.
Openvla_pred_42: " your action" and your action and your " your action" and your action and your
DriverGPT4_pred_42: 000000000000000000000000000000000000000000000
VLA4CD_image_pred_42: My action is straight-line distance from you, and why is because I am going straight.
My action is straight-line distance from you, and why is because I am going straight. My action is straight-line distance from you, and why is because I am going straight.
VLA4CD_language_pred_42: - -

---

Input_43: You can see that there are 2 cars. Their speed, straight-line distance from you, and angle in the direction your heading are respectively 5.07 2.4 m/s, 14.89 9.64 m, 14.18 179.72 °.You are now -0.029m laterally away from your driving route.
Question_43: What safe action is suggested based on the current scenario?
VLA4CD_pred_43: What safe action is suggested based on the current scenario? What safe action is suggested based on the current scenario? What safe action is suggested based
Openvla_pred_43: What safe action is based on the current? What safe action is based on the current
DriverGPT4_pred_43: 000000000000000000000000000000000000000000000
VLA4CD_image_pred_43: straight-line distance from you, and angle in the direction your heading are respectively 5.07 2.4 m/s, 14.89 9.64 m, 14.18 179.72 °.
VLA4CD_language_pred_43: - - - ,- - ,-

---

Input_44: You can see that there are 2 cars. Their speed, straight-line distance from you, and angle in the direction your heading are respectively 4.98 2.38 m/s, 14.15 9.66 m, 14.95 179.71 °.You are now -0.032m laterally away from your driving route.
Question_44: Is the detected car in motion?
VLA4CD_pred_44: Yes  answer: No  answer: Yes  answer: No  answer

Openvla_pred_44: motion in the detected car in motion in the detected car in motion in the detected car in motion in the detected car in motion in the detected car in motion in the detected car in motion in the detected car in motion in the detected car in motion in the detected car in motion in the detected car in motion in the

DriverGPT4_pred_44: 000000000000000000000000000000000000

VLA4CD_image_pred_44: Yes, the detected car is in motion. ''''''''''''''''''''''''''''''''''''''''''

VLA4CD_language_pred_44: - -

