# OpenReview forum: "How to Build a Pre-trained Multimodal model for Simultaneously Chatting and Decision-making?"
_ICLR.cc/2025/Conference — ICLR 2025 Conference Withdrawn Submission_

### Official Review · Reviewer_vhvA · 2024-11-02

**Soundness:** 1
**Presentation:** 2
**Contribution:** 2
**Rating:** 5
**Confidence:** 3

**Summary:**

This paper opposed to simulating MLLMs as human in a real-world situation that require both chatting and decision making. For example, human drive can drive safely while having conversations with passengers. This is an important application problem in autonomous driving system.

**Strengths:**

1. The problem that developing a chatting and simultanenous decision making, itself is underexplored and important.
2. The proposed model gives both reasonable question answering output and reasonable action output.

**Weaknesses:**

1. I don't buy the idea that by simply concatenating the question answer data and action prediction data together and supervised finetune the LLaVA-like MLLM can solve the proposed problem. As described in the abstract, driving and chatting simultaneously, however the proposed LLM-based model is autoregressive decoding. From the architecture overview, we can see this proposed model can only provide a prediction after a very long answer output. No inference speedup or simultaneous decoding technology is being used or proposed to achieve this.
2. The only contribution to me seems combining action prediction and question answer data, which is very trivial. No siginificant improvement is achieved compared with specialist model in each single task. And the approach to combine these two tasks chatting and decision making tasks are not actually achieving prediction both but sequentially.
3. The paper neeeds to be revised in the writing. Such as notions in Figure 1 is totally missing. Training and architecture details are missing in Experiments section, etc.

**Questions:**

1. What is the average number of images per training and inference case?

---

> ### Author Response · Authors · 2024-11-15
> **Response to Weaknesses and Questions**
>
> Thanks for your time and efforts in reviewing our paper! We highly appreciate your thoughtful and constructive suggestions. Your thoughtful and constructive suggestions have been invaluable to us, and we have carefully considered each comment. Our responses to your queries are outlined below:
>
> ***Response to Weaknesses1:***
>
> Thank you very much for your recognition of our proposed questions, but I would also like to take this opportunity to provide a more detailed introduction to our motivation. As you mentioned, "As stated in the abstract, driving and chatting simultaneously, however, the proposed LLM-based model is autoregressive decoding. From the architecture overview, it can be seen that the model can only provide predictions after a very long answer output." The motivation behind the VLA4CD model is to simulate the human ability to handle multiple tasks in complex environments, particularly in the parallel processing of action decision-making and dialogue generation. With the rapid advancements in LLMs within the field of NLP, significant research has focused on fine-tuning pre-trained LLMs to perform various tasks in specific domains. Existing approaches often involve fine-tuning LLMs independently for each task using its respective training data. This approach not only incurs high computational costs but also isolates the knowledge of each task, preventing efficient sharing across tasks, which can lead to suboptimal overall performance.
>
> To address these issues, we propose the VLA4CD integrated model. This model consolidates data from all tasks and performs unified fine-tuning on a single pre-trained LLM, while also incorporating specialized loss functions tailored for multimodal and multitask learning. This design significantly reduces computational overhead and, through efficient task sharing, enhances the overall performance of the VLA model in multitask scenarios.
>
> ***Response to Weaknesses2:***
>
> We propose the VLA4CD integrated model. This model consolidates data from all tasks and performs unified fine-tuning on a single pre-trained LLM, while also incorporating specialized loss functions tailored for multimodal and multitask learning. This design significantly reduces computational overhead and, through efficient task sharing, enhances the overall performance of the VLA model in multitask scenarios.
>
> Your point that "the method of combining these two tasks (chatting and decision-making) does not actually predict simultaneously but sequentially" is very insightful. Due to our unclear expression, it caused a misunderstanding. VLA4CD emphasizes not only multimodal output but also multimodal synergy, specifically sequential prediction. In autonomous driving tasks, the importance of decision-making is higher than question answering. Therefore, we can enhance decision-making through text and images, as shown in Table 4, where we maintain the input as images and text, and removing the language part from the loss function significantly reduces the system's decision-making ability, corresponding to line 483, "we see that including language in the loss function significantly enhances the quality of decision-making." This means sequentially predicting A based on text sensor input and Q, and then predicting action based on images, text sensor input, and Q together. This sequential prediction highlights the priority and importance of the two tasks, which we believe is a very valuable point.
>
> ***Response to Weaknesses3:***
>
> Thank you for your detailed prompts. We will thoroughly check the errors in the paper and revise and explain the points you mentioned regarding the interpretation of our views.
>
> ***Response to Questions1:***
>
> During training, the minimum batch size for images is 8, with a context length of 1 or 4 in each batch. During inference, real-time inference is performed for each frame image based on the environment.
>
> ***In summary***
>
> Overall, you have raised very valuable and insightful comments. We will ensure to revise the paper and clearly address the points you mentioned. Finally, we sincerely hope that these explanations will alleviate your concerns, and we sincerely hope that you will reconsider your score.

---

### Official Review · Reviewer_UzGf · 2024-11-03

**Soundness:** 3
**Presentation:** 4
**Contribution:** 2
**Rating:** 3
**Confidence:** 5

**Summary:**

- The authors propose VLA4CD, a model for self-driving that is capable of generating both language and action tokens.
- VLA4CD is trained using three different types of loss functions: language generation, action generation, and image reconstruction.
- The trained model demonstrates superior performance in both decision-making and question answering compared to models such as DriverGPT4 and OpenVLA in the gym-carla environment.

**Strengths:**

The main contribution of this work is that the authors add a language generation capability to VLA in self-driving scenarios.
- The paper is well-structured, making it easy to read and understand the authors' approach.
- The finding that separating language and action prediction loss can improve decision-making is a significant contribution that provides valuable insights into how VLAs can be effectively trained. It is encouraging to see that this is empirically demonstrated to be useful in self-driving scenarios. However, it is concerning that introducing some language noise into the training dataset can have a considerable impact on decision-making processes. Since real-world datasets will inevitably contain substantial noise, developing methods to ensure robustness against such noise is essential for the model's practical application.

**Weaknesses:**

- I don't quite understand why VLAs need to chat based on the author's motivation. Chatting is an inherently multi-turn conversation with a specific topic, but example of such capability of the model is completely lacking. I wonder what the authors' definition of chatting is. The model doesn't actually "chat" but simply outputs action description. It is far from the example in the introduction where authors want to build a model that can talk with a friend while driving.
- Text generation has already been explored with DriveGPT4. In this paper, text generation is not used for any novel applications other than simply translating action tokens into language. I fail to understand why does the author claim text+action generation is something novel since there's already a model that does it.
- Adding chat capabilities could potentially make the model less robust when exposed to noise in language interactions. Since the model learns to associate language with specific action tokens, any slight disruption to that association (e.g., due to noise) could significantly impair its action prediction performance. If the model can engage in unrestricted conversation, it is likely to encounter more noise, which could seriously affect its decision-making abilities, which is the most important goal of VLAs. It might be more effective for VLAs to focus solely on action prediction and incorporate chat functionality with separate models. With the current motivation, it seems there is no strong rationale or necessity to integrate chat capabilities into VLAs.
- Following the point above, I feel like the paper could be better framed on how to manage loss when training VLAs, which is a much interesting topic.

Typo: Figure 1 interatcion

**Questions:**

- How does the model compare to DriveGPT4 and why does DriveGPT4 do so bad? DriveGPT4 is doing exactly same thing as this model aims to do (text generation + action generation).
- Why are there no use case of the model actually chatting? How do the authors define chatting? The example of the introduction mentions the authors are inspired by human driver talking to a friend while driving, but the model doesn't actually engage in free form chat that goes beyond a single step.
- How does author plan to make the model robust to noise when exposed to unrestricted chat?
- Why should we add language generation capability to VLAs? The motivation for that seems non-existent in this paper and there's no novel use case of the generated language.
- Why do the authors think using separate loss for language and action generation (unlike DriveGPT4) improves the decision making performance?
- Why do the authors only focus on the self-driving task?

---

> ### Author Response · Authors · 2024-11-15
> **Response Part (1/3)**
>
> Thanks for your time and efforts in reviewing our paper! We highly appreciate your thoughtful and constructive suggestions. Your thoughtful and constructive suggestions have been invaluable to us, and we have carefully considered each comment. Our responses to your queries are outlined below:
>
> ***Response to Weaknesses1: "What is the motivation of our paper?"***
>
> The motivation behind the VLA4CD model is to simulate the human ability to handle multiple tasks in complex environments, particularly in the parallel processing of action decision-making and dialogue generation. With the rapid advancements in LLMs within the field of NLP, significant research has focused on fine-tuning pre-trained LLMs to perform various tasks in specific domains. Existing approaches often involve fine-tuning LLMs independently for each task using its respective training data. This approach not only incurs high computational costs but also isolates the knowledge of each task, preventing efficient sharing across tasks, which can lead to suboptimal overall performance.
>
> To address these issues, we propose the VLA4CD integrated model. This model consolidates data from all tasks and performs unified fine-tuning on a single pre-trained LLM, while also incorporating specialized loss functions tailored for multimodal and multitask learning. This design significantly reduces computational overhead and, through efficient task sharing, enhances the overall performance of the VLA model in multitask scenarios.
>
> ***Response to Weaknesses 2 and 3: Please refer to the answers below for Q1, Q2, and Q3.***
>
>
> ***Response to Q1: How does the model compare to DriveGPT4 and why does DriveGPT4 do so bad? DriveGPT4 is doing exactly same thing as this model aims to do (text generation + action generation).***
>
> Compared to DriveGPT4, VLA4CD demonstrates significant advantages, particularly in multitask parallel processing and output effectiveness.
>
> Firstly, DriveGPT4 is designed with a multimodal input but single-modal output framework, generating both text and action instructions through a detokenizer. However, this approach has inherent limitations, especially in generating fine-grained decision instructions at each moment. This is because text generation and action generation tasks in DriveGPT4 are not independent but share the same generation process. This leads to task conflicts, particularly in complex scenarios, where the model cannot simultaneously generate efficient text and actions. As shown in Figure 2 and Tables 1, 2, and 3, due to this task conflict, DriveGPT4 cannot consistently generate efficient and complete text and action instructions, severely affecting its decision accuracy and dialogue capabilities.
>
> In contrast, VLA4CD employs a multimodal input and multimodal output architecture, with separate objective functions specifically for text generation and action generation. This ensures that the model can generate efficient text and action instructions simultaneously when handling complex tasks. By designing independent objective functions, VLA4CD can output text and actions in parallel, avoiding task conflicts. This parallel processing not only simplifies system design but also significantly enhances the model's decision-making ability and efficiency in multitask complex environments.
>
> Moreover, VLA4CD's parallel processing mode effectively improves the model's adaptability in multi-objective tasks, ensuring the effectiveness and completeness of text generation and action decision-making. In contrast, DriveGPT4, due to its shared generation mechanism, often struggles with task conflicts, resulting in the inability to generate high-quality text and precise action instructions simultaneously. This parallel design makes VLA4CD far superior to DriveGPT4 in multi-objective tasks.

---

> ### Author Response · Authors · 2024-11-15
> **Response Part (2/3)**
>
> ***Response to Q2: Why are there no use case of the model actually chatting? How do the authors define chatting? The example of the introduction mentions the authors are inspired by human driver talking to a friend while driving, but the model doesn't actually engage in free form chat that goes beyond a single step.***
>
> The motivation behind the VLA4CD model is to simulate the human ability to handle multiple tasks in complex environments, particularly in the parallel processing of action decision-making and dialogue generation. With the rapid advancements in LLMs within the field of NLP, significant research has focused on fine-tuning pre-trained LLMs to perform various tasks in specific domains. Existing approaches often involve fine-tuning LLMs independently for each task using its respective training data. This approach not only incurs high computational costs but also isolates the knowledge of each task, preventing efficient sharing across tasks, which can lead to suboptimal overall performance.
>
> To address these issues, we propose the VLA4CD integrated model. This model consolidates data from all tasks and performs unified fine-tuning on a single pre-trained LLM, while also incorporating specialized loss functions tailored for multimodal and multitask learning. This design significantly reduces computational overhead and, through efficient task sharing, enhances the overall performance of the VLA model in multitask scenarios.
>
> We define "chatting" as open-ended, multi-turn conversations, and VLA4CD enhances its parallel generation capabilities in decision-making and dialogue tasks through LoRA fine-tuning. Our model retains the inherent language generation and everyday chatting capabilities of large language models, although we did not demonstrate this in the paper. In the paper, we only showcased dialogues related to driving scenarios, as large language models inherently possess everyday chatting abilities.
>
> ***Response to Q3: How does author plan to make the model robust to noise when exposed to unrestricted chat?***
>
> Great question! The key points you mentioned are how the model maintains robust decision-making capabilities in the presence of increased noise and how to extend VLA4CD's everyday chatting abilities by altering the QA system content.
>
> As seen in Table 5, when sensor input noise increases, the model's decision-making capabilities are indeed affected. This is because noise interferes with the model's perception input, thereby affecting its decisions based on these inputs. However, VLA4CD is designed to minimize this impact when sensor inputs are relevant to driving tasks. Therefore, as long as the sensor input information is noise-free, the model can maintain strong robustness, ensuring that decision-making capabilities are not disrupted.
>
> For the need to extend everyday chatting capabilities, we can follow this design approach. When expanding the QA system, we can keep the sensor input relevant to the driving scenario and only adjust the QA dataset content. This approach does not interfere with the model's decision-making ability, as modifying the QA content does not involve changes to sensor data or environmental perception data. Therefore, by reasonably adjusting and expanding the QA pairs, the model can maintain efficient decision-making capabilities in driving tasks and exhibit stronger dialogue capabilities in extended chatting scenarios.
>
> In short, as long as the sensor input is effective and stable, the VLA4CD model can maintain high robustness and efficiency in decision tasks while adding everyday chatting functions.
>
> ***Response to Q4: Why should we add language generation capability to VLAs? The motivation for that seems non-existent in this paper and there's no novel use case of the generated language.***
>
> The VLA4CD model mimics human multitasking abilities in complex environments, aiming to generate multiple outputs simultaneously to enhance decision efficiency and user experience. Unlike traditional single-task models, VLA4CD handles both action decision-making and language generation tasks, simplifying system architecture and reducing coordination issues. By integrating language generation capabilities into a unified model, VLA4CD better adapts to complex tasks, improving decision consistency and effectiveness. Inspired by human ability to simultaneously make decisions and converse in autonomous driving scenarios, VLA4CD was initially designed for specific tasks but innovates by extending multitask processing capabilities, enabling the model to handle a broader range of applications, enhancing adaptability and interactivity. For example, the VLA4CD model can manage multiple tasks in smart home management, such as adjusting temperature, controlling lighting, and checking security devices; it can also parallel process tasks in scenarios like controlling multiple robotic arm systems, showcasing its advantages in multitask parallel processing.

---

> ### Author Response · Authors · 2024-11-15
> **Response Part (3/3)**
>
> ***Response to Q5: Why do the authors think using separate loss for language and action generation (unlike DriveGPT4) improves the decision making performance?***
>
> We believe that using separate loss functions for language and action generation (unlike DriveGPT4) can improve decision performance mainly because this design avoids task conflicts and enables effective parallel processing of multiple tasks. In DriveGPT4, text generation and action generation share the same generation process, leading to conflicts between the two tasks, especially in complex environments where the model cannot simultaneously generate efficient text and action instructions. As a result, DriveGPT4 cannot guarantee fine-grained action instructions at each moment, affecting its decision accuracy and dialogue capabilities.
>
> In contrast, VLA4CD sets up independent objective functions for text generation and action generation, ensuring that each task can be processed efficiently in parallel, avoiding task conflicts. This design allows VLA4CD to generate text and action instructions simultaneously in complex tasks, thereby enhancing the model's decision-making ability and efficiency in multitask environments. Through independent loss functions, VLA4CD ensures the effectiveness and completeness of each output when handling multi-objective tasks, improving the model's adaptability and decision performance.
>
> ***Response to Q6: Why do the authors only focus on the self-driving task?***
>
> The question you raised is very important and indeed worth in-depth discussion. Our current research primarily focuses on autonomous driving tasks because we built the multimodal fusion, alignment, and system loss functions from scratch, fine-tuning a large model based on Llama-7B, and completed dataset collection, model establishment, and full evaluation and inference design. This process consumed a significant amount of time and cost, so we decided to first implement and evaluate the model in the autonomous driving scenario.
>
> However, this does not mean that we are limited to the autonomous driving field. In the future, we fully intend to extend this approach to other areas such as robotic arms and achieve multiple objective task outputs (e.g., parallel processing of three or more tasks). Autonomous driving is merely the starting point for verifying and testing this multitask processing framework. As the technology matures and the model is optimized, we plan to promote it to broader application scenarios.
>
> ***In summary***
>
> Overall, you have raised very valuable and insightful comments. We will ensure to revise the paper and clearly address the points you mentioned. Finally, we sincerely hope that these explanations will alleviate your concerns, and we sincerely hope that you will reconsider your score.

---

> ### Comment · Reviewer_UzGf · 2024-11-26
>
> W1: I understand that using 2 large models for chatting and decision making is computationally costly, however, that is not a strong motivation for combining two since this operates on an assumption that you need a on-board chatting LLM. Furthermore, if authors want to prove that multitask training improves driving performance, I recommend comparing it with non-chatting baselines on standard metrics like reviewer RHJw suggested. In the end, a self-driving LLM is of no practical use if it's core action generation capability lags behind other baselines. Chatting is just an auxiliary feature.
>
> Q1: Makes sense, thanks!
>
> Q2: I agree with the definition of chatting, can the authors provide a metric or evaluation that proves that the VLA4CD retains it's chatting capability? There are works that show some mix of instruction tuning data harms general conversational capability (e.g. https://aclanthology.org/2024.acl-srw.15.pdf).
>
> Q3: It makes sense but I don't think any of that is empirically shown in the paper?
>
> Q4: Same concern as W1.
>
> Q5&6: Makes sense.

---

> ### Comment · Reviewer_UzGf · 2024-11-26
>
> Further concerns:
>
> W1: How do you ensure chat ability is not affected by your finetuning? There's no empirical evidence on that and there are cases where general open-ended dialogue capability gets harmed by finetuning on specific downstream tasks (look at my comment above for reference).
>
> W2: How do you ensure that the system is robust toward noise in the dialogue? Since open-ended dialogue inherently will contain scenarios unrelated to driving, it needs to be robust toward any type of dialogue but the paper does not show that.
>
> W3: You mentioned that "We did not use the CARLA leaderboard method as a baseline," and failed to explain why you did not do that other than explaining your metric. I am still skeptical of why authors chose a difficult path of implementing their own metrics while you could easily compare the model with public baseline using the same environment.

---

### Official Review · Reviewer_FHJw · 2024-11-05

**Soundness:** 1
**Presentation:** 2
**Contribution:** 1
**Rating:** 3
**Confidence:** 3

**Summary:**

The current manuscript proposes to build a large language model (LLM) capable of understanding multiple modalities like text, vision, and actions; and producing them as outputs. In particular, it develops a Visual Language Action model for Chatting and Decision Making (VLA4CD) that produces continuous actions without losing its ability to chat with a user simultaneously. Notably, the action space is not discretized and kept continuous, unlike prior works in this area. The paper also demonstrates experiments on CARLA dataset to claim that this approach is effective and can provide real-time decision making compared to prior art, while retaining its text interaction capability.

**Strengths:**

(S1) In general, the intended direction of this work, i.e., a model that can take actions while retaining the ability to generate textual responses to a user is useful. Please see weaknesses for further discussion.

(S2) The technical details as presented in the paper are easy to understand and follow.

**Weaknesses:**

(W1) The current manuscript suffers from a clear lack of motivation for why we need a model that can produce both actions and also “chat” (L21, for instance) with a user. There are two main problems here:
* Throughout, the ability of “chatting with people” (L88) has not been characterized well. It is not open-ended dialog on any topic but rather an explanation of what actions to take or why it has taken a certain action in a given situation. This is misleading as currently phrased.
* Much of the motivation is around “a human driver can operate while chatting with a friend”, which does not apply to why we need a unified model. For instance, why not have an actuation model and an open-ended dialog model in the autonomous vehicle to achieve the above desired motivation? This indicates the lack of a clear motivation from an application standpoint.


(W2) Even if one were to scope the “chatting with users” ability down to producing explanations as responses to a fixed set of templated questions (see A.10), the manuscript does not follow through via corresponding experiments. Both actions and text-generation capability has been evaluated independently, once again begging the question as to why such a unified system is useful. There are no experiments to verify the following:
* The model actually actuates based on the textual outputs? I.e., if the model responds with “I will take the right lane in 20 mins”, does it actually do that?
* Are these textual explanations correct/sound given the state of the environment?
* What is the correlation of the GPT-4o score evaluation with human evaluation?


(W3) There are some concerns around the experimental validation of the proposed methodology:
* The reported experiments on town03 and town04 from the CARLA environment do not seem to match with any of the existing benchmarks with prior works (C, D).
* To further exacerbate this issue, none of the baseline results are from literature and have been reported based on reproductions in this work.
* Missing baselines, see [A] for more information.
This raises serious questions about the efficacy and usefulness of the proposed methods from an empirical standpoint. Why were existing, standardized benchmarks not used for model comparisons? Request the authors to address these concerns without which the benefits of this approach will remain unclear.


References
* [A] DriveMLM: Aligning Multi-Modal Large Language Models with Behavioral Planning States for Autonomous Driving. https://arxiv.org/pdf/2312.09245.
* [B] Think2Drive: Efficient Reinforcement Learning by Thinking with Latent World Model for Autonomous Driving (in CARLA-v2). https://arxiv.org/pdf/2402.16720
* [C] CARLA Autonomous Driving Leaderboard. https://leaderboard.carla.org/
* [D] TransFuser: Imitation with Transformer-Based Sensor Fusion for Autonomous Driving. https://arxiv.org/pdf/2205.15997

**Questions:**

(Q1) L154: Do you also include model textual response \hat{w}_i, i = {1,..,H} in w_i?

(Q2) Eq 1: \hat{w} is overloaded as both the model textual response and the embeddings of text inputs

(Q3) Eq 1: Each of w_i might have a different number of tokens (different from n). Do you pad them to n or is the index (n_i) dropped for brevity? That is: (w_i^1, w_i^2….w_i^{n_i}) instead of just (w_i^1, w_i^2….w_i^{n})

(Q4) L222-L225: The observed phenomenon is not clear here. Referring to the appendix also doesn’t add more details, apart from the empirical observation. Can the authors describe this with an example?

(Q5) VLA4CD (no-language): What is the architecture, inputs for this model? Ideally, the human question in the input and the {s_t^{l+1}, .., s_t^{l+n}} must be removed while training.

(Q6) L413: How did you balance the two losses for DriverGPT4? Did you have a hyperparameter search for the loss weights similar to your approach?

(Q7) L477: The reasoning here is heavily dependent on the discretization strategy used for each environment. How were the actions discretized for this environment? Was there a hyperparameter search performed to get the best strategy?

(Q8) L140-142: How is this problem avoided in the current setup? It’s not clear in the text here.
* Action space dimension is small, i.e., 2 (acceleration and steering) How does this scale with more variables?

---

> ### Author Response · Authors · 2024-11-15
> **Response  to Weaknesses**
>
> Thanks for your time and efforts in reviewing our paper! We highly appreciate your thoughtful and constructive suggestions. Your thoughtful and constructive suggestions have been invaluable to us, and we have carefully considered each comment. Our responses to your queries are outlined below:
>
> ***Response to Weaknesses1: "What is the motivation of our paper?"***
>
> The motivation behind the VLA4CD model is to simulate the human ability to handle multiple tasks in complex environments, particularly in the parallel processing of action decision-making and dialogue generation. With the rapid advancements in LLMs within the field of NLP, significant research has focused on fine-tuning pre-trained LLMs to perform various tasks in specific domains. Existing approaches often involve fine-tuning LLMs independently for each task using its respective training data. This approach not only incurs high computational costs but also isolates the knowledge of each task, preventing efficient sharing across tasks, which can lead to suboptimal overall performance.
>
> To address these issues, we propose the VLA4CD integrated model. This model consolidates data from all tasks and performs unified fine-tuning on a single pre-trained LLM, while also incorporating specialized loss functions tailored for multimodal and multitask learning. This design significantly reduces computational overhead and, through efficient task sharing, enhances the overall performance of the VLA model in multitask scenarios.
>
> We define "chatting" as open-ended, multi-turn conversations, and VLA4CD enhances its parallel generation capabilities in decision-making and dialogue tasks through LoRA fine-tuning. Our model retains the inherent language generation and everyday chatting capabilities of large language models, although we did not demonstrate this in the paper. In the paper, we only showcased dialogues related to driving scenarios, as large language models inherently possess everyday chatting abilities.
>
> ***Response to Weaknesses2:***
>
> In both generating dialogues and action generation capabilities, we generate them simultaneously, as shown in Figure 5. The independent evaluation of text and action generation capabilities is solely to demonstrate the comparison between VLA4CD and models with only single dialogue or decision-making abilities. The model does not actually execute actions based on text outputs; as shown in Figure 5(a), we unify the action outputs for decision control to maintain consistency. Regarding text explanations, we use GPT-4o for scoring, as shown in Figure 4, where VLA4CD outperforms other solutions in this scenario, but cannot guarantee 100% correctness. Since GPT-4o has strong language understanding capabilities and avoids human evaluator subjectivity, we chose the third-party GPT-4o for scoring, which is also a mainstream evaluation method currently[1][2][3].
>
> [1] Fu, Jinlan, et al. "Gptscore: Evaluate as you desire." arXiv preprint arXiv:2302.04166 (2023).
>
> [2] Geng, Xinyang, et al. "Koala: A dialogue model for academic research." Blog post, April 1 (2023): 6.
>
> [3] Sun, Zhiqing, et al. "Principle-driven self-alignment of language models from scratch with minimal human supervision." NeurIPS 2024.
>
> ***Response to Weaknesses3:***
>
> We use benchmarks based on the Gym-Carla environment, designed specifically for reinforcement learning agents in autonomous driving scenarios. Since the core of reinforcement learning is optimization through reward functions, we customized the evaluation matrix according to task requirements, as detailed in Appendix A.5, which defines multiple evaluation criteria. These criteria may differ from traditional benchmarks in the CARLA leaderboard, leading to differences in experimental results. We use these customized evaluation matrices because our dataset is collected based on reinforcement learning agents, whose behaviors and adaptability need to be evaluated through reinforcement learning-specific methods. To better assess the explicit feedback mechanism and continuous learning capabilities we propose, these customized matrices can more accurately reflect the actual effectiveness of the method.

---

> ### Author Response · Authors · 2024-11-15
> **Response to Questions**
>
> ***Response to Q1 and Q2:***
>
> You have pointed out a very good issue. I apologize for the oversight in our writing. The entire lines 151-157 regarding the character description should be revised as follows:
>
> > "We consider a multimodal setting similar to (Xiao et al., 2020), wherein, at each time step $t$, upon the agent performing an action  $a_t$, the environment returns an observation consisting of both visual and textual modalities, denoted by $(o_t, \hat{w}_t)$. Our objective is to build a generative model:
> >
> > $\pi$(a_t, \hat{w*}_t | o_t-H, \hat{w}_t-H, a_t-H, ....... , o_t, \hat{w}_t)
> >
> > which can generate both high-quality action decisions and text responses, given a sequence of historic observations and actions. Here, $\hat{w*}_t $ denotes a text-formed response to the text-formed input $\hat{w}_t$. If $\hat{w}_t$  is a question, then  $\hat{w*}_t $ can be seen as its answer given by our model.  $H$ denotes the length of the context."
>
> ***Response to Q3:***
>
> For simplicity, we omitted the index (n_i). We set the token counts for text and image inputs uniformly to 424 and 64, respectively, corresponding to the parameter `cutoff_len` in Table 6 and `num_patches` in Table 7.
>
> ***Response to Q4:***
>
> As shown in the different time series descriptions in Appendix 11, the text descriptions of Other Sensors Input 32 to Other Sensors Input 44 mainly come from fixed templates, with little variability, almost solely dependent on numerical changes. In real scenarios, this variability would significantly increase. Using the cross-entropy loss function may cause the model to struggle distinguishing between different time-step Other Sensors Input descriptions, leading to inaccurate predictions. As a result, the final output action values are almost identical, insensitive to numerical changes in text inputs, and unable to achieve finer-grained decision-making and differentiation. Therefore, adding label smoothing loss can enhance the model's perception of differences in Other Sensors Input descriptions at different time steps, making it more sensitive to numbers, and enabling finer-grained decision-making and control.
>
> ***Response to Q5:***
>
> Although the VLA4CD (no language) model architecture is still based on Figure 1, with inputs still being images and text, ideally, human questions and {s_t^{l+1}, .., s_t^{l+n}} should be removed during training. However, to quickly verify the impact of removing language loss, we adopted the direct removal approach.
>
> ***Response to Q6:***
>
> When balancing the two losses for DriverGPT4 and OpenVLA, we used the same hyperparameter settings as VLA4CD, with the specific parameter settings detailed on lines 263-264.
>
> ***Response to Q7:***
>
> We adopted the same discretization strategy for `action_bin` as OpenVLA and RT2, which involves discretizing continuous actions (acceleration and steering) into a fixed number of bins using a uniform discretization method. The discretized action values are then mapped to the end tokens of the pre-trained tokenizer's vocabulary to generate the corresponding tokens for actions. Similarly, these tokens can be inversely mapped back to the original continuous action values. This strategy allows encoding continuous actions as discrete tokens while preserving the precision of the continuous action information.
>
> ***Response to Q8:***
>
> This is a very good question. In Table 9, we compared the designs with and without image reconstruction loss when the inputs are both images and text. We found that the design with image reconstruction loss further enhances the final decision control. Our action space extends to multiple dimensions, which can be fully achieved by changing the output dimension of the action linear projection layer. Here, we only use two dimensions because we mainly consider the two key variables in autonomous driving scenarios.
>
> ***In summary***
>
> Overall, you have raised very valuable and insightful comments. We will ensure to revise the paper and clearly address the points you mentioned. Finally, we sincerely hope that these explanations will alleviate your concerns, and we sincerely hope that you will reconsider your score.

---

> ### Comment · Reviewer_UzGf · 2024-11-26
>
> I moved my comment under my review to avoid confusion. Sorry for the mistake!

---

> > ### Author Response · Authors · 2024-11-26
> >
> > Thank you very much for your feedback.
> >
> > Thank you for your responses regarding W1, Q2, Q3, and Q4. Please refer to the Post-Rebuttal responses to reviewer FHJw for W1, W2, and W3 for detailed information.

---

> ### Comment · Reviewer_FHJw · 2024-11-26
> **Post-Rebuttal**
>
> Huge thanks to the authors for duly responding to my concerns about the current work.
>
> Post the author comments, the current standings of the three weaknesses that I initially raised are as follows:
>
> (W1) **Unclear motivation:**
>
> The authors' response "Our model retains the inherent language generation and everyday chatting capabilities of large language models, although we did not demonstrate this in the paper." is a claim that is ungrounded empirically. If the authors propose to use the original weights of the language model (without LoRA weight offsets) for open-ended dialog (thus retaining its abilities), then the novelty of this contribution is further diminished as it simply demonstrates LoRA's ability and not that of the current work.
>
> (W2) **Relatedness of text responses to action outputs, evaluation:**
>
> As the authors' respond that the textual outputs to the questions are *independent* of the actions, this seriously questions the motivation further. In essence, a model can get away with generating response that look-like text without being grounded in the current state or any semblance of soundness. This is a huge concern wrt application of this work.
>
> (W3) **No comparisons to performance numbers from literature**:
>
> All quantitative numbers reported are the authors' experiments on customized settings within CARLA environment, without comparisons to numbers from prior works (despite pointing out some options in the review), thus raising questions around the soundness of empirical comparisons. Author's response that reinforcement learning requires customized benchmarks is not valid as other related works do consider consistent comparisons to systematically benchmark progress (see review).
>
> As a result, I'd like to maintain the current ratings and encourage the authors to more empirically address these weaknesses in further iterations of this work.

---

> ### Author Response · Authors · 2024-11-26
>
> Thank you very much for your feedback.
>
> ***Response to W1:*** Thank you for your valuable suggestions. We can further clarify our motivation. Most existing models follow a 1+1 multimodal pattern, while our VLA4CD framework implements a 1+2 multimodal fine-tuning pattern in LLMs, which can be similarly extended to a 1+N multimodal fine-tuning pattern. Here, the first "1" represents the input of one modality, and the second "2" or "N" represents the output of two or N modalities. Our motivation is to fully utilize the information contained in one input modality to output multiple modalities while ensuring that the output modalities have the capability of their corresponding 1+1 models. Additionally, our experimental results show that VLA4CD performs exceptionally well in both QA and decision-making capabilities for specific scenarios compared to existing methods. Therefore, the proposed 1+2 fine-tuning approach can improve the utilization and decision-making efficiency of the model's input modalities, and it can be further extended to a 1+N pattern in the future. Furthermore, since the large model itself already possesses basic everyday chatting capabilities, there is no need to fine-tune the QA capability for everyday chatting through LoRA. Therefore, to differentiate the content from everyday conversational abilities, we chose task-specific QA for specific scenarios, aiming to distinguish it from the inherent chatting content of the original model and demonstrate the effectiveness of our method in expanding multitask capabilities.
>
> ***Response to W2:*** Thank you for your valuable thoughts. We emphasize that the textual QA output is independent of actions for the following reasons: The input to our model is sensor input + random Q + image, where Q is randomly sampled from the question set in Appendix A.10, which includes questions related to daily driving scenarios and also empty Qs. Since Q is generated randomly at each time step, but VLA4CD demonstrates strong decision-making capabilities while answering random questions, as shown in Table 1, 2, and 3, this indicates that the decision-making capability is independent of the QA content, which also suggests that the QA content can be further expanded. However, it is important to note that the sensor input must be relevant to the current driving scenario, while the content of Q can be random. This can be seen in Table 5, where adding noise to the sensor input affects the final decision-making capability. Therefore, as long as the information in the sensor input is noise-free and the content of QA is expandable, the model can maintain strong robustness and ensure that the decision-making capability is not disturbed.
>
> ***Response to W3:*** Thank you for your concern. We did not use the CARLA leaderboard method as a baseline. Instead, we proposed two main metrics: Driving Score (DS) and Average Reward (AR), where DS = ER * AR, as detailed in Appendix A.5. Therefore, DS is strongly correlated with AR. The calculation of AR is based on the same reward function $f$ [1] from the gym-carla Benchmark, similar to most RL evaluation methods. Our VLA4CD also uses the same reward function to evaluate different baselines' AR, which demonstrates the rationality of using AR for evaluation. The reward function in Appendix A.4 includes comprehensive evaluations of collision behavior, lateral speed control, longitudinal speed control, overspeed control, lane keeping, steering and driving stability, and static punishment. We added ER because, although AR covers most scenario evaluations, the static punishment in the original function f is only "-0.1," which is insufficient. AR alone cannot consider the scenario of standing still. In the gym-carla Benchmark, the entire episode is reset if any collision or lane departure occurs. Therefore, we introduced DS = ER * AR, where ER = N_completed steps / N_total steps, representing the effective driving completion rate without penalty round interruption mechanisms. Thus, DS ensures evaluation under effective driving conditions, including more reasonable AR scores under static conditions. DS can be seen as a discount on AR scores, indicating that our evaluation is reasonable. From Table 1, Table 2, and Table 3, VLA4CD's AR and DS values are leading, which I believe is sufficient to demonstrate the effectiveness of our method.
>
> [1].Chen, Jianyu, Shengbo Eben Li, and Masayoshi Tomizuka. "Interpretable end-to-end urban autonomous driving with latent deep reinforcement learning." IEEE Transactions on Intelligent Transportation Systems 23.6 (2021): 5068-5078.

---

### Note · Authors · 2024-11-29

I have read and agree with the venue's withdrawal policy on behalf of myself and my co-authors.